# A Mutual Information Perspective on Multiple Latent Variable Generative Models for Positive View Generation

**Dario Serez**                                                                                     *dario.serez@iit.it*
*Istituto Italiano di Tecnologia, Italy*

**Marco Cristani**                                                                        *marco.cristani@univr.it*
*Reykjavík University, Iceland*
*University of Verona, Italy*

**Alessio Del Bue**                                                                           *alessio.delbue@iit.it*
*Istituto Italiano di Tecnologia, Italy*

**Vittorio Murino**                                                                         *vittorio.murino@iit.it*
*Istituto Italiano di Tecnologia, Italy*
*University of Verona, Italy*

**Pietro Morerio**                                                                         *pietro.morerio@iit.it*
*Istituto Italiano di Tecnologia, Italy*

**Reviewed on OpenReview:** *https://openreview.net/forum?id=uaj8ZL2PtK*

## Abstract

In image generation, Multiple Latent Variable Generative Models (MLVGMs) employ multiple latent variables to gradually shape the final images, from global characteristics to finer and local details (*e.g.*, StyleGAN, NVAE), emerging as powerful tools for diverse applications. Yet their generative dynamics remain only empirically observed, without a systematic understanding of each latent variable's impact. In this work, we propose a novel framework that quantifies the contribution of each latent variable using Mutual Information (MI) as a metric. Our analysis reveals that current MLVGMs often underutilize some latent variables, and provides actionable insights for their use in downstream applications.

With this foundation, we introduce a method for generating synthetic data for Self-Supervised Contrastive Representation Learning (SSCRL). By leveraging the hierarchical and disentangled variables of MLVGMs, our approach produces diverse and semantically meaningful views without the need for real image data. Additionally, we introduce a Continuous Sampling (CS) strategy, where the generator dynamically creates new samples during SSCRL training, greatly increasing data variability. Our comprehensive experiments demonstrate the effectiveness of these contributions, showing that MLVGMs' generated views compete on par with or even surpass views generated from real data.

This work establishes a principled approach to understanding and exploiting MLVGMs, advancing both generative modeling and self-supervised learning. Code and pre-trained models at: https://github.com/SerezD/mi_ml_gen.

## 1 Introduction

*Latent Variable Generative Models* (LVGMs), such as Variational Autoencoders (VAEs) (Kingma & Welling, 2014; Rezende et al., 2014) and Generative Adversarial Networks (GANs) (Goodfellow et al., 2014), are foundational approaches for image generation. Given a random variable $\mathbf{X} \in \mathcal{X}$, representing high-dimensional

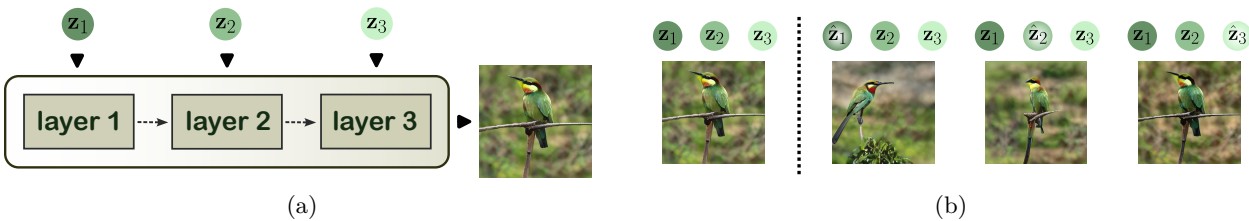

(a)                     (b)

Figure 1: **(a)** Multiple Latent Variable Generative Models utilize multiple latent variables (here $\mathbf{Z}_1, \mathbf{Z}_2, \mathbf{Z}_3$), which are sequentially introduced at different layers of the generative network to produce high-quality images. **(b)** The base image (left), generated using points $\mathbf{z}_1, \mathbf{z}_2, \mathbf{z}_3$, can be selectively modified by altering individual latents ($\mathbf{z}_1$ to $\hat{\mathbf{z}}_1$, $\mathbf{z}_2$ to $\hat{\mathbf{z}}_2$, or $\mathbf{z}_3$ to $\hat{\mathbf{z}}_3$). Each latent affects the final image differently, at first influencing broader, global attributes and later refining finer, local details (darker to lighter shading in the figure).

pictures in pixel space, LVGMs aim to approximate the underlying data distribution $p(\mathbf{X})$. To achieve this, they learn a parameterized generator $g(\mathbf{z}; \theta) = \mathbf{x}$, where $\mathbf{Z} \in \mathcal{Z}$ denotes a latent variable sampled from a simpler and known distribution in the $\mathcal{Z}$ latent space. A key objective of the learning process is to ensure that the generator is continuous, such that neighboring latent points $\mathbf{z}'$ and $\mathbf{z}''$ are mapped to perceptually similar outputs $\mathbf{x}'$ and $\mathbf{x}''$. This regularization of the latent space allows LVGMs to generate novel content and meaningfully interpolate latent features (Radford et al., 2016; Higgins et al., 2017).

Over the years, advancements in latent generative modeling have focused on the use of *multiple* latent variables, rather than a single latent code (Vahdat & Kautz, 2020; Karras et al., 2019; 2020; 2021; Sauer et al., 2022). By incorporating latent variables at different layers of the network (see Figure 1a), these Multiple Latent Variable Generative Models (MLVGMs) offer a hierarchical structure where early latent codes influence broad, global features and later codes refine finer, local details (Figure 1b). The resulting architecture enhances control over the image synthesis process, enabling the generation of high-resolution images with richer detail and improved precision.

The ability to disentangle global and local features in image generation not only improves the visual quality of generated images but also widens the application scope of these models. For instance, the StyleGAN architecture (Karras et al., 2019) has demonstrated exceptional performance in image editing (Alaluf et al., 2022; Pehlivan et al., 2023), manipulation (Tov et al., 2021), and translation (Richardson et al., 2021). Additionally, recent studies have shown that MLVGMs can serve as effective foundation models for tasks such as adversarial purification (Serez et al., 2025). Collectively, these findings highlight the versatility of MLVGMs, showcasing their utility not only in creative and generative domains but also as pre-trained models for broader applications.

Nevertheless, existing research primarily leverages the "global-to-local" behavior of MLVGMs as an empirical tool, applying it across diverse tasks without delving into the mechanics of latent variable utilization. In other terms, these approaches assume that earlier latent variables shape coarse image attributes while later ones refine fine details, but they do so without formally analyzing how each latent variable contributes to image generation. As a result, the internal dynamics of MLVGMs remain poorly understood.

To address this gap, we propose a novel approach that establishes a direct relationship between feature distances in each latent space ($\mathbf{Z}_1, \mathbf{Z}_2, \ldots, \mathbf{Z}_n$) and mutual information (MI) shifts in the shared image space $\mathbf{X}$. The key insight is that producing an equivalent MI shift in the output space requires increasingly larger perturbations ($\mu_i$) as we move deeper into the generative hierarchy—i.e., from $\mathbf{Z}_1$ to $\mathbf{Z}_n$. This quantitatively confirms the global-to-local pattern and reveals how influence diminishes across successive latent variables (see Figure 2).

By grounding this analysis in information theory, our approach moves beyond intuition and provides a principled framework for understanding latent variable roles in MLVGMs. It enables direct comparisons between latent spaces and exposes inefficiencies in how models allocate representational power. Notably, we reveal the underutilization of later-stage latents in all tested models, opening to further architectural improvements and enabling better strategies for leveraging MLVGMs in downstream tasks.

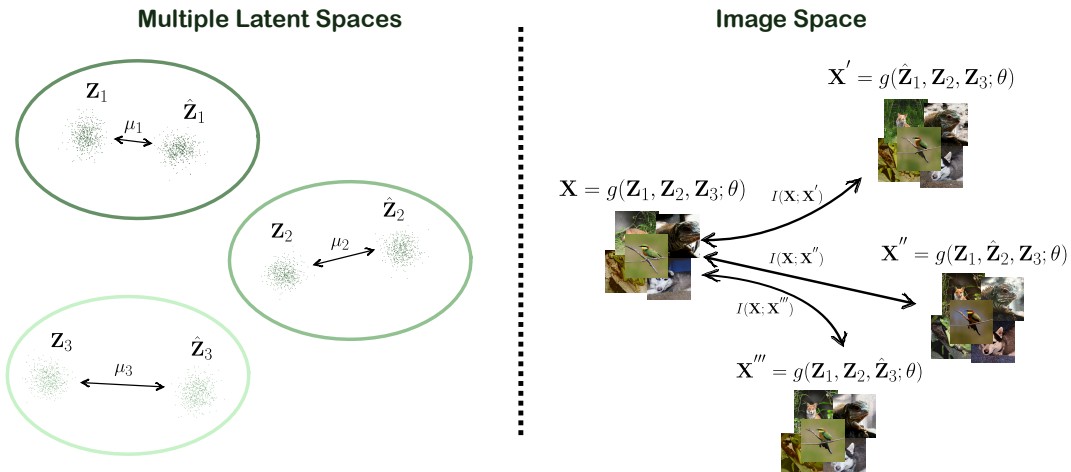

Figure 2: Illustration of our findings on the "global-to-local" behavior in MLVGMs. On the left, perturbations are applied to each latent variable independently ($\mathbf{Z}_1 \to \hat{\mathbf{Z}}_1$, $\mathbf{Z}_2 \to \hat{\mathbf{Z}}_2$, and $\mathbf{Z}_3 \to \hat{\mathbf{Z}}_3$), with the average perturbation magnitude increasing across latent spaces ($\mu_1 < \mu_2 < \mu_3$). On the right, each perturbed latent variable is used to generate modified images ($\mathbf{X}'$, $\mathbf{X}''$, and $\mathbf{X}'''$) from the original $\mathbf{X}$. Notably, the increasing perturbation magnitude in the latent space maintains approximately equal Mutual Information shifts in the image space: $I(\mathbf{X}; \mathbf{X}') \approx I(\mathbf{X}; \mathbf{X}'') \approx I(\mathbf{X}; \mathbf{X}''')$. This provides the first quantitative measure of the "global-to-local" property, where earlier latents affect global features and later latents refine local details.

With this understanding, we propose a novel application of MLVGMs in Self-Supervised Contrastive Representation Learning (SSCRL). In SSCRL, feature extractors, or encoders $f(\mathbf{x}; \phi)$ with parameters $\phi$, are trained to represent data by contrasting positive and negative views. Positive views are semantically similar images, encouraged to have close representations in the latent space, while negative views correspond to unrelated data points that are forced to have distant representations. Therefore, we propose to leverage the different impacts of multiple latent variables in MLVGMs to manipulate specific features and generate positive views. This approach enables the training of SSCRL encoders *without relying on real data*, demonstrating the potential of MLVGMs as pre-trained models for producing high-quality synthetic images tailored for representation learning.

The primary objective of SSCRL is to enforce a desired set of invariances in the learned representations (Xiao et al., 2020), achieved by creating valid positive views. Figure 3 compares the proposed method with standard pixel-space augmentations and single latent variable generative models (LVGMs) for view generation. In the typical approach (Figure 3a), a finite set of hand-crafted transformations, such as color adjustments, cropping, or flipping, is applied directly in the pixel space. Alternatively, invariances can be introduced at the latent level of a pre-trained LVGM (Figure 3b). However, in LVGMs, all image features are entangled within a single latent space, making it difficult to generate specific invariances (*e.g.*, altering fur patterns) without inadvertently affecting global features, such as changing the dog breed (*e.g.*, from Australian Terrier to Yorkshire Terrier in the figure). In contrast, MLVGMs inherently disentangle global and local features, enabling precise control over specific characteristics. For instance, using MLVGMs, attributes like fur patterns or color can be modified while preserving global features, such as the dog breed. This is achieved by independently perturbing each latent variable to a desired magnitude, as illustrated in Figure 3c.

The use of generative models to create both anchor and positive views introduces a significant challenge: the lower classification accuracy typically observed when training on synthetic data compared to real data (Ravuri & Vinyals, 2019). Prior studies, such as Besnier et al. (2020); Lampis et al. (2023), have identified the lack of diversity in generated images as a primary factor contributing to this issue. To mitigate this, these works propose increasing dataset diversity by sampling and storing a large number of synthetic images before training. In contrast, we propose a novel approach called *Continuous Sampling* (CS) to address this limitation. With CS, new images are generated "online" during the encoder network's training process, offering three key advantages: (i) no need to store large quantities of synthetic data, thereby preserving disk space; (ii) no data

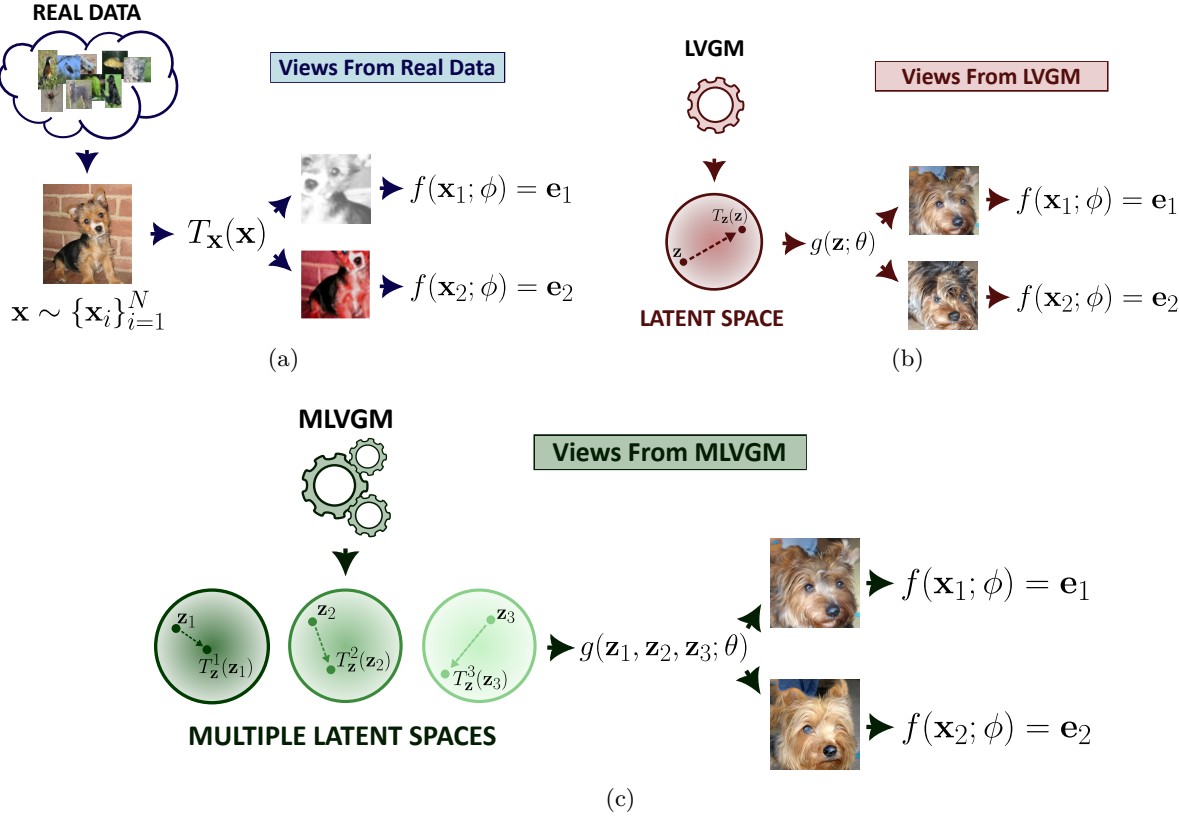

Figure 3: Self-Supervised Contrastive Representation Learning (SSCRL) optimizes an embedding function $f(\mathbf{x}; \phi) = \mathbf{e}$, mapping semantically similar images $\mathbf{x}_1, \mathbf{x}_2$ to nearby latent representations $\mathbf{e}_1, \mathbf{e}_2$. **(a)** In the classic approach, positive views are generated by applying hand-crafted transformations in the pixel space, $T_{\mathbf{x}}$, to a finite dataset of images $\mathbf{x} = \{\mathbf{x}_i\}_{i=1}^N$. **(b)** Alternatively, positive views can be generated by sampling nearby points in the latent space of a Latent Variable Generative Model (LVGM), $g(\mathbf{z}; \theta)$. However, since image features remain highly entangled in the latent space, even subtle perturbations $T_{\mathbf{z}}(\mathbf{z})$ may change important characteristics such as dog breed (Australian to Yorkshire Terrier in the figure). **(c)** Our framework leverages a Multiple Latent Variable Generative Model (MLVGM), represented as $g(\mathbf{z}_1, \mathbf{z}_2, \ldots, \mathbf{z}_n; \theta)$ ($n = 3$ in the figure). By applying tailored perturbations $T_{\mathbf{z}}^i(\mathbf{z}_i)$ to each latent variable, we leverage the hierarchical feature representation to obtain a broader range of valid transformations while maintaining important semantic aspects.

loading step, which is often the bottleneck in neural network training, as new batches are generated directly into GPU memory; and (iii) maximized diversity, specifically by ensuring that each batch is freshly sampled and never reused, unlike prior methods that rely on a fixed-size synthetic dataset.

To evaluate our contributions, we apply our novel quantification algorithm to two distinct MLVGMs: a BigBiGAN (Donahue & Simonyan, 2019) pre-trained on ImageNet-1K (Deng et al., 2009), and a StyleGAN2 (Karras et al., 2020) pre-trained on LSUN Cars (Yu et al., 2015). Subsequently, we leverage the same MLVGMs to generate views for SSCRL using our proposed Continuous Sampling (CS) strategy. Specifically, we train feature extractors with three different SSCRL frameworks, namely SimCLR (Chen et al., 2020), SimSiam (Chen & He, 2021), and BYOL (Grill et al., 2020). The learned representations are then validated following a consolidated practice through linear classification across multiple downstream datasets and object detection on Pascal VOC (Everingham et al., 2010). Our results demonstrate that MLVGM-based view generation outperforms simple LVGM-based techniques and *achieves comparable or superior results than training with real data*. Additionally, we measure the training time per epoch using Continuous Sampling *vs.* standard data loading, establishing CS as an efficient alternative for increasing data diversity.

To sum up, our contributions are as follows: i) We propose the first method to quantify the influence of individual latent variables in Multiple Latent Variable Generative Models (MLVGMs), which can reveal underutilized latent spaces and serve as a helpful tool for downstream applications. ii) We leverage the natural disentanglement of coarse from fine features in MLVGMs to create positive views for Self-Supervised Contrastive Representation Learning (SSCRL), enabling tailored invariances that outperform previous methods using both real and synthetic data. iii) We introduce Continuous Sampling, a novel procedure that dynamically generates new batches during SSCRL training, increasing data diversity, reducing storage requirements, and maintaining competitive training time performance.

## 2 Related Works

**MLVGMs.** The idea of utilizing multiple latent variables is well-established in the generative models' literature, typically presented as an evolution of Latent Variable Generative Models (LVGMs). For instance, Variational Autoencoders (VAEs) (Kingma & Welling, 2014; Rezende et al., 2014) leverage multiple latent variables to enhance the expressivity of approximate distributions, as demonstrated by architectures such as NVAE (Vahdat & Kautz, 2020) and VD-VAE (Child, 2021), or to improve latent disentanglement, as in Li et al. (2019). Similarly, Generative Adversarial Networks (GANs) (Goodfellow et al., 2014) have embraced this concept in models like LapGAN (Denton et al., 2015), BigGANs (Brock et al., 2018; Donahue & Simonyan, 2019), and GigaGAN (Kang et al., 2023). Advances in Normalizing Flows (Dinh et al., 2015; Rezende & Mohamed, 2015) have also incorporated multiple latent variables, with works like Hu et al. (2022) introducing architectures inspired by physics to achieve this goal.

In such a growing environment, numerous applications of MLVGMs have emerged. Of particular relevance in this context is the StyleGAN family (Karras et al., 2019; 2020; 2021; Sauer et al., 2022), which has been widely applied in image editing and manipulation tasks (Tov et al., 2021; Richardson et al., 2021; Alaluf et al., 2022; Pehlivan et al., 2023). More recently, MLVGMs have also been used as foundation models for non-generative downstream tasks, such as purification against adversarial attacks (Serez et al., 2025). Motivated by such a rapid expansion, in this work we propose a novel application in SSCRL and for the first time study how the "generative load" is spread across latent variables in the current architectures.

**SSCRL view generation.** Self-Supervised Contrastive Representation Learning (SSCRL) (Hadsell et al., 2006) aims to learn meaningful latent representations without relying on labeled data, primarily by designing informative positive views (Tian et al., 2020; Xiao et al., 2020). Early approaches, such as Bachman et al. (2019); Misra & Maaten (2020); Caron et al. (2020), focused on pretext tasks like matching global and local parts of an image to create multiple views. Subsequently, SimCLR (Chen et al., 2020), a foundational method in the field, introduced the use of manually designed transformations, including flipping, cropping, and color distortions. More recent works have explored advanced techniques, such as learning views in an adversarial manner (Tamkin et al., 2020; Shi et al., 2022) or projecting anchor images into the latent spaces of pre-trained generators (Yang et al., 2022; Astolfi et al., 2023; Kim et al., 2023; Wu et al., 2023; Han et al., 2023; Zeng et al., 2024). While our approach differs by relying solely on synthetic data, we show that it is also complementary to previous techniques, allowing to apply different transformations (denoted as $T_{\mathbf{x}}(\mathbf{x})$ in Figure 3a) on top of the generated views-*i.e.* directly in the pixel space.

Further along our line of work, methods like Jahanian et al. (2021); Li et al. (2022) have proposed generating fully synthetic views by sampling nearby points in the latent space of pre-trained LVGMs (Figure 3b). We direct compare against these baselines in the experimental section, showing that the coarse-to-fine feature disentanglement of MLVGMs allows to obtain better views for most downstream tasks.

Finally, recent efforts have explored the generation of synthetic views in a text-to-image setting (Tian et al., 2024a;b). While this direction holds promise, particularly when combined with MLVGMs, its application to our framework remains limited. This is primarily due to the lack of publicly available code and pre-trained models for text-to-image MLVGMs, such as GigaGAN (Kang et al., 2023).

**Training with generated data.** The remarkable performance of modern generative models, such as Rombach et al. (2022); Chang et al. (2023), has opened up new possibilities for using synthetic data to train

classifier networks. A common strategy involves augmenting real datasets with generated samples, which has shown promise in enhancing classification performance (He et al., 2022; Bansal & Grover, 2023; Azizi et al., 2023). Alternatively, more ambitious efforts attempt to train classifiers entirely on synthetic data, leveraging advanced text-to-image models to obtain high-quality datasets (Sariyildiz et al., 2023; Singh et al., 2024).

The primary challenge in these approaches is the limited diversity of generated data, which has been identified as a key factor contributing to the performance gap between classifiers trained on real versus synthetic datasets (Ravuri & Vinyals, 2019). Recent studies (Fan et al., 2024) suggest that scaling up the size of synthetic training sets can reduce this accuracy gap, though it does not fully eliminate it. However, generating large datasets introduces its own set of challenges, particularly increased disk space usage and data management overhead. Existing methods (Besnier et al., 2020; Lampis et al., 2023) address this issue by partially renewing synthetic data at each epoch or by regenerating the dataset entirely every $N$ epochs. In this work, we surpass the common assumption about the inefficiencies of generating training images in real time, leveraging fast-sampling models, such as GANs, to generate data directly during training and avoiding storage and loading bottlenecks altogether.

## 3 Methodology

### 3.1 Measuring the impact of latent variables in MLVGMs

Before formalizing our approach for measuring the contribution of single latent variables, we define the concept of Multiple Latent Variable Generative Models (MLVGMs):

**Definition 1** (Multiple Latent Variable Generative Models).

*A Multiple Latent Variable Generative Model (MLVGM), denoted $g(\mathbf{z}_1, \mathbf{z}_2, \ldots, \mathbf{z}_n; \theta) = \mathbf{x}$, is a deep neural network parameterized by $\theta$. It generates new data $\mathbf{x}$ by modeling $n$ random latent variables $\{\mathbf{z}_1, \mathbf{z}_2, \ldots, \mathbf{z}_n\}$ at different and progressive layers of the network, such that:*

$$g : \mathbb{R}^{m_1} \times \mathbb{R}^{m_2} \times \cdots \times \mathbb{R}^{m_n} \to \mathbb{R}^d$$
$$g := l_{[n]}(\mathbf{z}_n, l_{[n-1]}(\mathbf{z}_{n-1}, \ldots l_{[1]}(\mathbf{z}_1) \ldots));$$

*where $l_{[i]}$ represents the $i^{th}$ block of the generator, and $\mathbf{z}_i$ is the corresponding latent variable at that layer (parameters $\theta$ are omitted for clarity).*

**Intuitions.** To meaningfully compare the contribution of each latent variable $\mathbf{Z}_i$, we need a metric that operates in the common pixel-space. Probabilistically, we consider images as a random variable $\mathbf{X}$, and therefore

select *Mutual Information (MI)* as the metric of choice[1].

As an example, let's consider an MLVGM with $n = 3$ latent variables, as shown in Figure 1. Let $\mathbf{Z}_1, \mathbf{Z}_2, \mathbf{Z}_3$ represent the random latent variables for the three latent spaces, and $\mathbf{X}$ the output in the pixel space. Suppose we perturb the first latent variable, replacing $\mathbf{Z}_1$ with $\hat{\mathbf{Z}}_1$. This generates a modified random variable $\mathbf{X}'$ in the pixel space. We can now relate the average magnitude of the perturbation in the latent space (*e.g.* using L$_2$ distance), $\mu_1 = \mathbb{E}[\|\hat{\mathbf{Z}}_1 - \mathbf{Z}_1\|_2]$, to the resulting Mutual Information shift in the pixel space, $I(\mathbf{X}, \mathbf{X}') = \gamma$.

The same process can be repeated for $\mathbf{Z}_2$ and $\mathbf{Z}_3$, introducing $\hat{\mathbf{Z}}_2$ and $\hat{\mathbf{Z}}_3$, and calculating the perturbation magnitudes $\mu_2$ and $\mu_3$ needed to achieve the *same* MI shift $\gamma$ in the pixel space. If the generative process respects the "global-to-local" hierarchy typically attributed to MLVGMs (Figure 1b), we expect: $\mu_3 > \mu_2 > \mu_1$, as depicted in Figure 2.

Since directly computing MI for high-dimensional variables like $\mathbf{X}$ is analytically intractable, we estimate a lower bound using InfoNCE (Oord et al., 2018). Additionally, we employ a Monte Carlo procedure to calculate the average perturbations. Details of these computations are provided in the following sections.

---

[1]See Appendix A for the formal definition of Mutual Information and its probabilistic interpretation.

**Preliminaries.** InfoNCE loss (Oord et al., 2018) was originally proposed for SSCRL, encouraging similar views (positives) to have close representations, while ensuring that dissimilar views (negatives) remain distant. Formally, it is defined as:

$$\mathcal{L}_{\text{InfoNCE}} = \mathbb{E}_{\mathbf{x}, \mathbf{x}'} \left[ -\log \left( \frac{e^{\text{sim}(f(\mathbf{x}; \phi), f(\mathbf{x}'; \phi))/\tau}}{\sum_{k=1}^{K} e^{\text{sim}(f(\mathbf{x}; \phi), f(\mathbf{x}^k; \phi))/\tau}} \right) \right]; \tag{1}$$

where $\mathbf{x}$ and $\mathbf{x}'$ are the anchor and positive images, respectively, sim denotes the cosine similarity operator, $f$ is the encoder function parameterized by $\phi$, $\tau$ is a temperature parameter and $K$ is the number of samples (both positive and negative) in a mini-batch.

As demonstrated in Oord et al. (2018); Poole et al. (2019), InfoNCE provides a lower bound on the MI between the learned representations:

$$\log(2K - 1) - \mathcal{L}_{\text{InfoNCE}} \leq I(f(\mathbf{X}; \phi); f(\mathbf{X}'; \phi)). \tag{2}$$

In typical SSCRL setups (e.g., SimCLR (Chen et al., 2020)), the random variables $\mathbf{X}$ and $\mathbf{X}'$ are generated using deterministic augmentations, such as cropping, flipping, or color adjustment, applied to the same base image. These transformations result in a fixed mutual information value $I(\mathbf{X}; \mathbf{X}')$. Since $f(\cdot; \phi)$ is a deterministic function, the fixed term $I(\mathbf{X}; \mathbf{X}')$ serves as an upper bound to Equation (2), following directly from the data processing inequality (see Appendix A):

$$\log(2K - 1) - \mathcal{L}_{\text{InfoNCE}} \leq I(f(\mathbf{X}; \phi); f(\mathbf{X}'; \phi)) \leq I(\mathbf{X}; \mathbf{X}'). \tag{3}$$

Thus, minimizing the InfoNCE loss in SSCRL can be interpreted as tightening the bounds on mutual information, ensuring that the learned representations effectively capture all relevant information shared between the positive views $\mathbf{X}$ and $\mathbf{X}'$.

**The proposed approach.** We build on the insights of Equation (3), utilizing InfoNCE as a proxy to measure MI shifts between views. Unlike classical SSCRL methods, which rely on fixed, deterministic transformations, we generate views $\mathbf{X}$ and $\mathbf{X}'$ by perturbing individual latent variables in the latent spaces of a pre-trained MLVGM.

Drawing inspiration from Li et al. (2022), which learns latent-space perturbations for positive view generation in LVGMs, our approach adopts an adversarial procedure to optimize InfoNCE loss while progressively reducing the MI between the positive views $\mathbf{X}$ and $\mathbf{X}'$[2].

Formally, let $g$ denote a pre-trained MLVGM with $n$ latent variables and parameters $\theta$. The objective is to identify a perturbation function $T_{\mathbf{z}}^i(\cdot)$ for each latent space $1 \leq i \leq n$, ensuring that:

$$I(g(\mathbf{Z}_1, \mathbf{Z}_2, \ldots, \mathbf{Z}_i, \ldots, \mathbf{Z}_n; \theta); g(\mathbf{Z}_1, \mathbf{Z}_2, \ldots, T_{\mathbf{z}}^i(\mathbf{Z}_i), \ldots, \mathbf{Z}_n; \theta)) \approx \gamma. \tag{4}$$

To achieve this, we model $T_{\mathbf{z}}^i(\mathbf{z}_i)$ as a simple additive perturbation: $T_{\mathbf{z}}^i(\mathbf{z}_i) = \mathbf{z}_i + p(\mathbf{z}_i; \varphi)$, where $p(\cdot)$ is a small multi-layer perceptron (MLP) parameterized by $\varphi$. Since InfoNCE provides the lower bound on MI, we need to compute it by introducing an encoder function $f(\cdot)$ with parameters $\phi$ and define the optimization as a minimax problem (we omit the parameters $\theta$ of the fixed generator $g$):

$$\max_{\varphi} \min_{\phi} \mathcal{L}_{\text{InfoNCE}}\big(f(g(\mathbf{z}_1, \mathbf{z}_2, \ldots, \mathbf{z}_i, \ldots, \mathbf{z}_n); \phi), f(g(\mathbf{z}_1, \mathbf{z}_2, \ldots, T_{\mathbf{z}}^i(\mathbf{z}_i; \varphi), \ldots, \mathbf{z}_n); \phi)\big); \tag{5}$$

---

[2]A detailed discussion of Li et al. (2022) is provided in Section 3.2.

**Training dynamics.** We initialize parameters $\varphi$ such that $T_{\mathbf{z}}^i(\cdot)$ represents the identity function. In other terms, the applied perturbation is initially zero, and the views $\mathbf{X}$ and $\mathbf{X}'$ are identical. From the perspective of Equation (3), $I(\mathbf{X}; \mathbf{X}') = H(\mathbf{X})$, corresponding to the trivial setting where the encoder $f$ can achieve $\mathcal{L}_{\mathrm{InfoNCE}} \approx 0$ with ease. As training continues, the perturbation function $T_{\mathbf{z}}^i(\cdot)$ learns to apply progressively larger modifications to the latent variable $\mathbf{Z}_i$, increasing the diversity of the generated views. This, in turn, reduces the mutual information $I(\mathbf{X}; \mathbf{X}')$, thereby *lowering the upper bound* in Equation (3). As a result, the encoder $f$, tasked with minimizing $\mathcal{L}_{\mathrm{InfoNCE}}$, must maintain the shared information between increasingly distinct views $\mathbf{X}$ and $\mathbf{X}'$ into a common representation, *tightening the lower bound.*

In summary, $T_{\mathbf{z}}^i$ progressively enhances diversity in the views, reducing $I(\mathbf{X}; \mathbf{X}')$ and causing InfoNCE to increase over time. Conversely, $f$ seeks to learn the most informative representations, tightening the lower bound from the left and seeking equality in Equation (3). We refer the reader to Appendix B for a detailed graphical illustration of these training dynamics, showing the evolution of InfoNCE loss, average perturbations, and additional insights such as required training time and hyperparameter settings.

**Monte Carlo sampling.** As a result of the above, we obtain $n$ independent perturbation functions $T_{\mathbf{z}_i}(\cdot)$, acting on single latents. For each of these, training is stopped when $\mathcal{L}_{\mathrm{InfoNCE}} \approx \overline{\gamma}$, ensuring views with a consistent MI shift across all latent variables, and allowing the direct comparison of perturbation magnitudes across different latent spaces.

Afterwards, we perform Monte Carlo (MC) sampling by computing a statistically relevant number of image pairs for each level $i$: $\mathbf{X} = g(\mathbf{Z}_1, \mathbf{Z}_2, \ldots, \mathbf{Z}_i, \ldots, \mathbf{Z}_n; \theta)$ and $\mathbf{X}' = g(\mathbf{Z}_1, \mathbf{Z}_2, \ldots, T_{\mathbf{z}}^i(\mathbf{Z}_i), \ldots, \mathbf{Z}_n; \theta)$, where we know that $I(\mathbf{X}; \mathbf{X}') \approx \overline{\gamma}$. This enables to estimate the *average latent perturbation* $\mu_i$ required to achieve a similar MI shift in the image space. As depicted in Figure 2, we generally expect that later latent spaces require larger perturbations to achieve the MI shift, matching the empirical observations on the "global-to-local" property of MLVGMs.

In Section 4, we use this strategy to estimate the impact of latent variables for two distinct MLVGMs: a BigBiGan Donahue & Simonyan (2019) pre-trained on ImageNet-1K Deng et al. (2009) and a StyleGan2 Karras et al. (2020) pre-trained on LSUN Cars Yu et al. (2015). The former has 6 latent variables, while the latter has 16, which we re-organize into 4 groups of 4 for computational practicality.

## 3.2 Positive view Generation Strategies

As illustrated in Figure 3c, we generate pairs of positive views by applying perturbations to one or more latent spaces, each with an appropriately selected magnitude. To better understand the rationale behind our approach, we first look at how previous methods decide view generation strategies.

**Background.** The problem of Self-Supervised Contrastive Representation Learning (SSCRL) is strictly correlated to designing effective positive views, enabling meaningful representations. In Tian et al. (2020), the following principle is introduced:

**Proposition 1** (Optimal Views for SSCRL, Tian et al. (2020)).
*Given a downstream task $\mathcal{T}$ with labels $\mathbf{Y} \in \mathcal{Y}$, the optimal views $(\mathbf{X_1^*}; \mathbf{X_2^*})$ created from data $\mathbf{X}$ are:*

$$(\mathbf{X_1^*}; \mathbf{X_2^*}) = \underset{\mathbf{X_1}; \mathbf{X_2}}{\arg\min}\, I(\mathbf{X_1}; \mathbf{X_2}); \; subject\ to\ I(\mathbf{X_1}; \mathbf{Y}) = I(\mathbf{X_2}; \mathbf{Y}) = I(\mathbf{X}; \mathbf{Y}); \tag{6}$$

*meaning that the Mutual Information (MI) between optimal views is minimized to contain only the task-relevant information $I(\mathbf{X_1^*}; \mathbf{X_2^*}) = I(\mathbf{X}; \mathbf{Y})$, while removing all nuisance information, $I(\mathbf{X_1^*}; \mathbf{X_2^*}|\mathbf{Y}) = 0$.*

The principle states that optimal views should minimize their Mutual Information (MI) while retaining all information relevant to the downstream task, expressed by some label $\mathbf{Y}$. However, in SSCRL, labels are unavailable, and *the downstream task is unknown.* Consequently, designing optimal views becomes infeasible. To solve this inherent shortcoming, most methods design views by implicitly fixing an MI threshold that decides true positives:

**Definition 2** (Mutual Information bound for SSCRL).
*Let $\mathbf{X}_1, \mathbf{X}_2$ be two random variables in the common image space. In absence of a known downstream task $\mathcal{T}$ with labels $\mathbf{Y} \in \mathcal{Y}$; the variables $\mathbf{X}_1, \mathbf{X}_2$ can be considered as positive views if*

$$I\big(\mathbf{X}_1, \mathbf{X}_2\big) \geq \rho; \tag{7}$$

*where $\rho$ is an implicitly defined **MI threshold**.*

In other terms, the goal of view generation is to define positives that share a *reasonable* amount of information, relevant for as many tasks as possible. To exemplify this phenomenon, we analyze three prominent positive view generation methods. SimCLR (Chen et al., 2020), defines a broad set of data augmentations $T_{\mathbf{x}}$ to be applied in the pixel space. The specific transformations and their combinations are selected through ablation studies conducted on the ImageNet-1K classification task. In this context, the threshold $\rho$ is implicitly fixed by means of Equation (6), where a pretext downstream task is considered.

In the context of Latent Variable Generative Models (LVGMs), two methods for generating views by perturbing the single latent space stand out. First, Jahanian et al. (2021) propose applying random perturbations to an anchor latent variable $\mathbf{z}$. Specifically, the perturbation is defined as $T_{\mathbf{z}}(\mathbf{z}) = \mathbf{z} + \mathbf{w}_{\text{rand}}$, where $\mathbf{w}_{\text{rand}} \sim \mathcal{N}^t(\mu, \sigma, t)$ follows a truncated Gaussian distribution with truncation parameter $t$. Similar to SimCLR, the parameters of the distribution (e.g., the standard deviation $\sigma$) are tuned via ablation studies on ImageNet, fixing $\rho$ with a pretext task.

In contrast, Li et al. (2022) introduce an adversarial approach to learn the perturbation $T_{\mathbf{z}}$ for each instance. In this case, the positive view is generated as $T_{\mathbf{z}}(\mathbf{z}) = \mathbf{z} + \mathbf{w}_{\text{learn}}$, where $\mathbf{w}_{\text{learn}}$ is a learnable perturbation vector. The objective function is formulated similarly to Equation (5), but applied to LVGMs. In this case, the training stopping criterion (and therefore the threshold $\rho$) is empirically decided by observing the quality of generated views at each step.

**Deciding perturbation magnitudes.** We maintain the perturbation strategies proposed by Jahanian et al. (2021) and Li et al. (2022), referred to as *random* and *learned*, respectively. In fact, the core advantage of our method relies on designing tailored magnitudes based on each latent space's contribution to the generative process, rather than proposing a novel perturbation strategy.

To do so, we proceed with two distinct phases. First, we use the quantification algorithm to relate the impact of different latents. This reveals the presence of over- or under-used codes, which we can immediately discard. Overused codes imply that even very small perturbations will result in a low $\rho$ threshold, while unused codes do not affect the threshold in any way. Therefore, this first step has the goal to discard perturbations that lead to non-informative views. Finally, in the second phase we consider the remaining codes only and visualize the semantic effect of the applied perturbations to estimate valid magnitudes, implicitly defining $\rho$.

As an example, in Figure 4 we generate multiple examples for each considered MLVGM, by perturbing each latent variable (or group, in the case of StyleGan2) with the same latent vector $\mathbf{w}$. For BigBiGan, the quantitative analysis reveals that the first code is overused, and we can verify that even small perturbations completely change the views semantic content. On the opposite, the last latent is underused, and visually presents no differences w.r.t. the reference image. As a result, we apply perturbations only on the remaining four latents, which act on small shape details or colors. Similarly, for StyleGan2, we first quantitatively verify that the last group is underused, and discard it. For the remaining ones, we visually observe how the first group controls large-scale transformations, the second group adjusts subject and background composition, while the latter is primarily responsible for color variations.

**Advantages.** The inherent disentanglement of coarse and fine-grained features in MLVGMs offers a clear advantage over standard LVGMs, even when employing similar perturbation strategies (*random* or *learned*). To clarify, consider the set of all possible downstream tasks for image data, $\{\mathcal{T}_1, \mathcal{T}_2, \ldots, \mathcal{T}_T\}$. For each latent point $\mathbf{z} \sim \mathcal{Z}$, there exists a maximum perturbation magnitude $|\mathbf{w}_{\mathbf{z}}|^{\mathcal{T}_t}$ for each task $\mathcal{T}_t$, such that the resulting views are optimal for the task (Proposition 1).

In SSCRL, however, the downstream task $\mathcal{T}_t$ is unknown. Latent perturbation-based positive view generation methods, therefore, aim to define a function $T_{\mathbf{z}}(\mathbf{z}) = |\mathbf{w}_{\mathbf{z}}|$ that estimates, for each latent point, a non-trivial

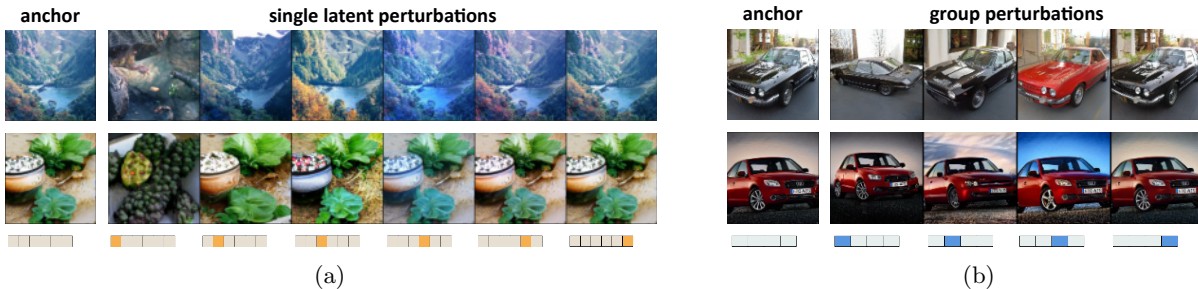

Figure 4: Examples of views generated by adding the **same** latent vector $\mathbf{w}$ to different levels. **(a)** Two anchor images and possible views generated by the perturbations of 6 BigBiGan's latent levels, represented as the 6 elements' vector at the bottom. The darker element indicates the applied perturbation $T_{\mathbf{z}}^i(\mathbf{z}_i) = \mathbf{z}_i + \mathbf{w}$ for each latent level $i$. **(b)** Generated anchors and views by StyleGan2, which has 16 hierarchical levels, grouped into 4 sets and represented as the 4 elements' vector at the bottom. The darker element indicates the altered group.

perturbation that generates valid (true) positives for as many tasks as possible. In other terms, it exists a direct mapping between the selected latent functions and the choice of the MI threshold $\rho$: while larger $|\mathbf{w}|$ (lower $\rho$) can yield more informative (hard) positives, it also increases the likelihood of producing false positives, potentially reducing generalization across diverse tasks.

The primary advantage of MLVGMs lies in their hierarchical structure, which distributes features across multiple latent spaces. This enables the definition of separate perturbation functions $T_{\mathbf{z}}^i(\cdot)$ for each latent space $i$, tailored to the impact of that space on the generative process. Crucially, the average perturbation magnitude $|\mathbf{w}|_i$ progressively increases with the latent level index $i$, as illustrated in Figure 3a. For example, perturbations in a high-level latent space (responsible for fine details like textures) can be very large without compromising the validity of positive views for most downstream tasks - *i.e.* they have a low effect on $\rho$.

Thus, for an MLVGM with $n$ latent spaces, perturbation magnitudes can be progressively scaled: $|\mathbf{w}|_1 < |\mathbf{w}|_2 < \cdots < |\mathbf{w}|_n$. In contrast, LVGMs encode all features within a single latent space, forcing perturbations to be constrained by the most sensitive features. From an MLVGM perspective, this corresponds to a uniform perturbation magnitude, $|\mathbf{w}|_1 = |\mathbf{w}|_2 = \cdots = |\mathbf{w}|_n$, which significantly limits flexibility and reduces the impact of generated positive views.

### 3.3 Continuous Sampling

Utilizing generative models to sample both anchor and positive views can degrade final performance (Ravuri & Vinyals, 2019), primarily due to the lower variability of synthetic images compared to real data. To address this limitation, previous methods (Besnier et al., 2020; Lampis et al., 2023; Fan et al., 2024) have proposed increasing variability by sampling a larger number of images relative to the reference dataset size, ensuring batches are not repeated across epochs. However, the prevailing approach involves sampling this extensive synthetic dataset *offline* (before training), which demands significant storage capacity and additional pre-processing time.

In this study, we avoid these drawbacks by adopting a *Continuous Sampling* strategy that leverages fast generators (such as GANs) to dynamically sample new images during the training of the SSCRL encoder. Specifically, we load the pre-trained generator onto the same GPU device as the encoder and replace the standard data loading step with an on-the-fly generator inference step. This process outputs a new batch of synthetic images directly on the target device, eliminating the need for pre-generated datasets. Since the pre-trained GAN operates exclusively in inference mode, the additional memory overhead is minimal and affordable, allowing us to maintain sufficiently large batch sizes for effective SSCRL training.

With this *Continuous Sampling* approach, the number of training steps per epoch remains consistent with real-data-based training. However, the total number of unique images seen during training is significantly increased, as the effective training set size becomes *n epochs* times larger. Moreover, this strategy eliminates

Table 1: Results of the MC simulation on BigBiGan **(a)** and StyleGan2 **(b)**. For each latent level $i$ or group $g$ we show the final InfoNCE loss value and the estimated mean ($\mu$) and standard deviation ($\sigma$) of the corresponding inferred distribution. Average perturbation values confirm that early levels greatly impact the generation process, while later levels may have no impact at all.

<table>
<tr><td align="center" colspan="4">(a)</td><td></td><td align="center" colspan="4">(b)</td></tr>
</table>

| latent level | loss | estimated $q^i$ | | | latent group | loss | estimated $q^g$ | |
|:---:|:---:|:---:|:---:|---|:---:|:---:|:---:|:---:|
| $(i)$ | (InfoNCE) | $(\mu_i)$ | $(\sigma_i)$ | | $(g)$ | (InfoNCE) | $(\mu_g)$ | $(\sigma_g)$ |
| 1 | 1.09 | 0.67 | 0.21 | | $1-4$ | 0.99 | 15.1 | 2.7 |
| 2 | 1.04 | 3.63 | 1.18 | | $5-8$ | 1.14 | 29.0 | 4.6 |
| 3 | 1.05 | 6.97 | 1.85 | | $9-12$ | 0.94 | 38.0 | 5.6 |
| 4 | 1.02 | 13.00 | 7.08 | | $13-16$ | 0.11 | 134.4 | 14.2 |
| 5 | 1.05 | 21.22 | 13.68 | | | | | |
| 6 | 0.14 | 594.71 | 616.80 | | | | | |

the need for pre-generating and storing extensive datasets and avoids standard data-loading bottlenecks, resulting in training times that are comparable to or faster than traditional methods (see Section 4). For a detailed implementation, we provide pseudocode for the continuous sampling procedure in Appendix C.

## 4 Experiments

In this section, we present the results of our Monte Carlo procedure for quantifying the impact of latent variables on two MLVGMs: BigBiGan and StyleGan2. Subsequently, we utilize these MLVGMs as view generators to train encoders using different SSCRL frameworks, leveraging our proposed Continuous Sampling strategy.

To evaluate the effectiveness of our approach, we compare it against two existing latent perturbation techniques for LVGMs, specifically those introduced by Jahanian et al. (2021) and Li et al. (2022). As an additional baseline, we include SimCLR, a widely-used view generation method applied to real data, and investigate its combination with transformations applied on top of MLVGM-generated views. Furthermore, in Appendix F, we extend the applicability of our method to other generative models beyond GANs by training an NVAE (Vahdat & Kautz, 2020) on the CIFAR-10 dataset (Krizhevsky et al., 2009).

Finally, we evaluate the overall training efficiency of Continuous Sampling by comparing its runtime performance against standard data loading pipelines, demonstrating its capability to increase data variability without incurring significant computational overhead.

### 4.1 Impact of Latent Variables

Following the procedure detailed in Section 3.1, we train $n$ separate perturbation functions $T_{\mathbf{z}}^i$ ($n = 6$ latent levels for BigBiGan and $n = 4$ latent groups for StyleGan2), optimizing the objective in Equation (5). As visually described in Appendix B, the InfoNCE loss rapidly decreases toward zero during the initial training iterations. As the perturbation functions $T_{\mathbf{z}}^i$ learn to apply increasing perturbations, the InfoNCE loss rises correspondingly. Training is terminated once a value of $\overline{\gamma} \approx 1$ is achieved, indicating an approximately equal MI shift in the pixel space.

For each latent level or group, we compute the learned perturbation $\mathbf{w_z} = p(\mathbf{z}; \varphi)$ across a statistically significant number of latent points $\mathbf{z}$. This enables us to estimate the mean ($\mu_i$ or $\mu_g$) and standard deviation ($\sigma_i$ or $\sigma_g$) of the inferred perturbation distributions $q^i(|\mathbf{w}|)$ or $q^g(|\mathbf{w}|)$. Table 1 presents these results, along with the final InfoNCE loss achieved during training.

From Table 1a (Monte Carlo results for BigBiGan), we observe that the average perturbation (estimated mean $\mu_i$) required to achieve a comparable InfoNCE loss increases progressively across latent levels, from $i = 1$ to $i = 5$. However, for $i = 6$, the InfoNCE loss does not rise substantially even under high average

Table 2: Comparison of representations learned on the ImageNet-1K dataset or BigBiGan generator with two contrastive frameworks (SimCLR and SimSiam). Metrics are Top-1 and Top-5 accuracy for linear classification on ImageNet-1K, average precision for detection on Pascal VOC, and mean Top-1 accuracy over 7 transfer classification datasets. "random" row refers to Jahanian et al. (2021), and "learned" to Li et al. (2022). **Bold** indicates the best result for each group, **underline** the absolute best, and $*$ indicates the baseline reported from Li et al. (2022).

| Data | $T_{\mathbf{z}}$ | $T_{\mathbf{x}}$ | SimCLR ImageNet-1K Top-1 | Top-5 | Pascal VOC AP | $AP_{50}$ | $AP_{75}$ | SimSiam ImageNet-1K Top-1 | Top-5 | Pascal VOC AP | $AP_{50}$ | $AP_{75}$ | Transfer Top-1 |
|---|---|---|---|---|---|---|---|---|---|---|---|---|---|
| real | - | all | **49.4*** | **75.6*** | **52.9*** | **78.7*** | **58.5*** | **49.1** | **74.2** | **54.4** | **80.0** | **60.0** | **58.2** |
| synth | - | all | 41.6* | 66.6* | 51.0* | 77.2* | 55.8* | 32.2 | 56.5 | 51.6 | 78.2 | 57.0 | 47.2 |
| synth | random | all | 48.7* | 73.1* | 50.2* | 77.0* | 54.4* | 33.4 | 57.7 | 51.7 | 78.4 | 56.3 | 47.0 |
| synth | ML rand. | no col. | **53.7** | **77.2** | **53.3** | **79.5** | **58.5** | **42.5** | **67.7** | **54.3** | **79.9** | **59.6** | **59.6** |
| synth | learned | all | 53.2* | 77.2* | 53.1* | 78.9* | 58.0* | 33.0 | 58.2 | 51.8 | 78.0 | 56.7 | 46.2 |
| synth | ML learn. | no col. | **54.4** | **77.9** | **53.4** | **79.5** | **58.9** | **39.5** | **64.8** | **52.5** | **78.9** | **57.5** | **54.9** |

Table 3: Comparison of representations learned on the LSUN Cars dataset or StyleGan 2 generator with two contrastive frameworks (SimSiam and Byol). Metrics are Top-1 and Top-5 accuracy for linear classification on Stanford Cars and FGVC Aircraft. "random" row refers to Jahanian et al. (2021), and "learned" to Li et al. (2022). **Bold** indicates the best result for each group, **underline** the absolute best.

| Data | $T_{\mathbf{z}}$ | $T_{\mathbf{x}}$ | SimSiam Stanford Cars Top-1 | Top-5 | FGVC Aircraft Top-1 | Top-5 | Byol Stanford Cars Top-1 | Top-5 | FGVC Aircraft Top-1 | Top-5 |
|---|---|---|---|---|---|---|---|---|---|---|
| real | - | all | **33.4** | **64.3** | 20.7 | 48.8 | **48.9** | **79.3** | **35.0** | **65.6** |
| synth | - | all | 27.0 | 54.6 | **21.3** | **50.5** | 40.5 | 69.6 | 31.2 | 61.9 |
| synth | random | all | 29.2 | 58.1 | 22.5 | 51.7 | 44.6 | 73.3 | 30.5 | 60.4 |
| synth | ML rand. | no col. | **47.0** | **76.1** | **22.9** | **53.5** | **58.7** | **84.8** | **32.5** | **61.8** |
| synth | learned | all | 28.6 | 56.7 | 22.0 | 51.9 | 45.6 | 73.6 | **31.7** | **62.1** |
| synth | ML learn. | no col. | **35.2** | **64.8** | **23.0** | **53.0** | **47.8** | **77.1** | 30.7 | 61.1 |

perturbations, suggesting an under-utilization of the latent level in the generative process. Conversely, we measure a very low $\mu_i$ for the first latent level, suggesting a possible over-utilization. These observations may indicate potential inefficiencies in the BigBiGan architecture or training procedure.

A similar trend is observed for StyleGan2 (Table 1b), where larger perturbation magnitudes ($\mu_g$) are needed to achieve comparable InfoNCE loss values as latent groups progress from $g = 1 - 4$ to $g = 13 - 16$. Notably, the final group exhibits a degenerate behavior, where even large perturbations fail to influence the MI of the generated views significantly.

Overall, these results provide clear quantitative evidence that the supposed global-to-local dynamics in MLVGMs hold. Specifically, early latent levels or groups exhibit a stronger influence on the generation process, while later ones primarily affect fine-grained details. To the best of our knowledge, this is the first empirical demonstration of such dynamics across MLVGMs.

### 4.2 View Generation

We test MLVGMs generated views by training multiple ResNet-50 encoders, using SimSiam (Chen & He, 2021), SimCLR (Chen et al., 2020) (on BigBiGan, following previous work Li et al. (2022)) and Byol (Grill et al., 2020) (on StyleGan2). We sample latent anchors from a truncated normal distribution: $\mathcal{N}^t(0.0, 1.0, 2.0)$

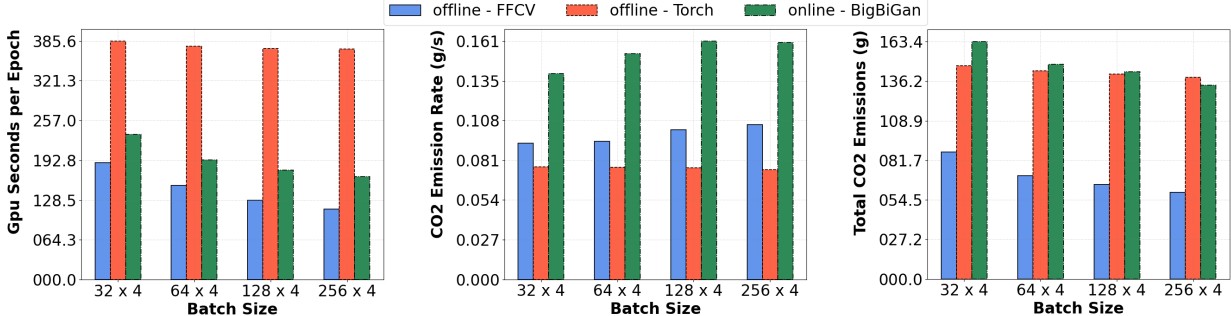

Figure 5: Total time (GPU seconds), $CO_2$ emissions rate (grams per second) and total $CO_2$ emissions (grams) for the three tested data loading procedures and different batch sizes.

for BigBiGan and $\mathcal{N}^t(0.0, 1.0, 0.9)$ for StyleGan2. Positives are computed using the *random* or *learned* strategies, applied separately on each latent level. Given the MC results reported in Table 1, we first select the latents to discard. These are the first latent of BigBiGan (overused) and the last latent/group in both BigBiGan and StyleGan2 (underused). Then, we systematically apply different perturbations to the remaining latents, observing that latents $2-5$ in BigBiGan obtain relevant semantic changes with a similar perturbation magnitude. For StyleGan2, we apply only tiny perturbations on the first two groups (modify shape and global transformations), and a larger perturbation on the remaining group (alters colors). The specific magnitudes, as well as other hyperparameters, are reported in Appendix D.

The representation capabilities of the obtained encoders are compared against several methods: training on synthetic data without latent perturbations $T_{\mathbf{z}}$, the *random* and *learned* baselines using single latent spaces, and the upper bound of using real data (1.28M images for ImageNet-1K (Deng et al., 2009) and 893K images for LSUN Cars (Yu et al., 2015)). In all these scenarios, SimCLR pixel-space augmentations $T_{\mathbf{x}}$ are used, consisting of random cropping, horizontal flipping, grayscale, and color jittering. Since our ML views generate realistic color changes (see Appendix H), we only partially apply $T_{\mathbf{x}}$ transformations on top of our positives, removing grayscale and color jittering. To better investigate this aspect, in Appendix E we further test various combinations of $T_{\mathbf{x}}$ coupled with our method.

BigBiGan views are evaluated on ImageNet-1K linear classification and, for Simsiam, on seven transfer datasets: Birdsnap (Berg et al., 2014), Caltech101 (Fei-Fei et al., 2004), Cifar100 (Krizhevsky et al., 2009), DTD (Cimpoi et al., 2014), Flowers102 (Nilsback & Zisserman, 2008), Food101 (Bossard et al., 2014), and Pets (Parkhi et al., 2012). We also compute Average Precision on Pascal VOC (Everingham et al., 2010) object detection using `detectron 2` (Wu et al., 2019) to train a Faster-RCNN with the R50-C4 backbone. The results are reported in Table 2, including the mean accuracy for the transfer tasks (complete runs in Appendix G). For StyleGan2, we compute linear classification accuracy on Stanford Cars (Krause et al., 2013) and FGCV Aircraft 2013b (Maji et al., 2013), reporting results in Table 3.

In all experiments, MLVGMs views outperform the corresponding baseline, proving their superior quality. Comparing *random* and *learned* methods, we observe that the multiple latent (ML) *random* experiments often close the gap with the *learned* counterparts. This suggests that distinct-level perturbations are more important than the selected alteration technique. In comparison with real data, ML views generally yield better or similar results, except in the case of SimSiam encoders evaluated on ImageNet-1K. However, this gap narrows or disappears in other downstream tasks and datasets, evidencing good generalization capabilities of the learned representations, which is the main goal of SSCRL. For StyleGan2, the great performance boost given by ML *random* views on Stanford Cars is noteworthy. When generalizing to FGCV Aircraft, all runs achieve similar performance, with marginal improvements of the ML runs when using SimSiam, and good real data results on Byol. This may be due to the high domain shift between the two datasets (Car vs Aircraft), leading to a challenging generalization for all representations.

### 4.3 Continuous Sampling

All our encoders are trained using Continuous Sampling, except for SimCLR, which follows the previous setup. Additionally, to compare overall training speed to standard data loading, we trained a ResNet-18 model with SimCLR for 20 epochs on ImageNet-100, on 4 NVIDIA A100-SXM4-40GB GPUs and different batch sizes ($32 \times 4$, $64 \times 4$, $128 \times 4$, $256 \times 4$). The experiment compares the standard `PyTorch` Paszke et al. (2019) loader, the efficient `FFCV` Leclerc et al. (2023) loader (both with 8 workers), and the BigBiGan generator. Figure 5 displays our findings, reporting the mean GPU seconds per epoch, the $CO_2$ emissions rate, and the total $CO_2$ emissions estimated using `CodeCarbon` Schmidt et al. (2021). Continuous Sampling proves significantly faster than `torch` loader and only marginally slower than `FFCV`. In terms of $CO_2$ emissions rate, the use of BigBiGan led to higher energy consumption, due to intensive GPU usage. Nevertheless, in terms of total $CO_2$ the values remain comparable with `torch`. In conclusion, the increasing efficiency and precision of modern image generation models, especially fast-sampling GANs, make Continuous Sampling an interesting alternative to conventional data-loading techniques, allowing great image variability while maintaining comparable training times.

## 5 Discussion and Conclusions

In this paper, we explored the influence of multiple latent spaces in MLVGMs' image generation, quantifying their impact as MI shifts in the common pixel space. This approach advances beyond previous empirical observations, providing deeper insights into the generative process, revealing under- or over-utilized latent variables, and guiding the use of MLVGMs in downstream applications. Additionally, we expanded the use of MLVGMs to a new downstream task, which is positive view generation for SSCRL, demonstrating superior results w.r.t. previous methods using single-variable models and competing with real data training. We also introduced Continuous Sampling, which allows using generators as a data source, creating large training sets without requiring significant storage capacity and achieving comparable or faster training times than standard data loading.

**Limitations and impact.** Our work showcases MLVGMs as a distinct category of models, offering new tools to assess the impact of latent variables. Specifically, the proposed Monte Carlo quantification method supports previous empirical observations on the "global-to-local" nature of MLVGMs, but allows a more in-depth and quantitative analysis. Nonetheless, our algorithm remains limited by the nature of Mutual Information itself, which is invariant under invertible transformations. As a consequence, some small perturbations (e.g. a constant scaling to each pixel), will be approximated as a 0 shift by our method. However, since the differences in average perturbation between each latent space are usually significant, it is reasonable to assume that such small approximations do not invalidate our findings. More specifically, we reveal that modern gan-based MLVGMs, such as BigGan and StyleGan employ over or under-utilized variables in the generative process, setting up a base for possible architectural improvements. In terms of view generation, our method has proven its superiority, surpassing previous perturbation strategies applied to single-variable models. However, it does not address the inherent challenge of SSCRL: views are defined upon "reasonable" thresholds, since "optimal" positives depend on the specific downstream task. Regarding generative models as a data source, they offer potential solutions to issues associated with real datasets, such as privacy concerns and usage rights Kaissis et al. (2020); DuMont Schütte et al. (2021). However, generative models can inherit biases from the original data Asim et al. (2020), and latent perturbations may amplify them, propagating to downstream models. Therefore, techniques to mitigate these biases could be considered Tan et al. (2020); Teo et al. (2023). As for Continuous Sampling, while it reduces disk usage and data diversity, scaling to high image resolutions or large models may be very GPU-intensive and result in high CO2 emissions, limiting usability.

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
