# A    Information Theory

**Mutual Information.**   Mutual Information (MI) measures the amount of information that one random variable contains about another. More specifically, it measures the reduction in the uncertainty of one random variable due to the knowledge of the other.

**Definition 3** *Mutual Information (MI).*

*Let $\mathbf{X}$ and $\mathbf{Y}$ be two random variables with joint probability $p(\mathbf{x}, \mathbf{y})$ and marginals $p(\mathbf{x})$ and $p(\mathbf{y})$. The Mutual Information (or MI in short) $I(\mathbf{X}; \mathbf{Y})$ is the Relative Entropy (or Kullback-Leibler divergence) between the joint distribution and the product of marginals distribution:*

$$I(\mathbf{X}; \mathbf{Y}) = D_{KL}(p(\mathbf{x}, \mathbf{y}) || p(\mathbf{x})p(\mathbf{y}))$$
$$= \int_{\mathbf{x} \in \mathcal{X}} \int_{\mathbf{y} \in \mathcal{Y}} p(\mathbf{x}, \mathbf{y}) \log \frac{p(\mathbf{x}, \mathbf{y})}{p(\mathbf{x})p(\mathbf{y})}.$$

Note that $I(\mathbf{X}; \mathbf{Y}) \geq 0$, with equality iff $\mathbf{X}$ and $\mathbf{Y}$ are conditionally independent.

**Data Processing Inequality.**    Formally, the data processing inequality can be formulated as:

**Theorem 1** *Data Processing Inequality.*

*If three random variables $\mathbf{X}, \mathbf{Y}, \mathbf{Z}$ form a Markov Chain ($\mathbf{X} \rightarrow \mathbf{Y} \rightarrow \mathbf{Z}$), then:*

$$I(\mathbf{X}; \mathbf{Y}) \geq I(\mathbf{X}; \mathbf{Z}),$$

*with equality iff $I(\mathbf{X}; \mathbf{Y}|\mathbf{Z}) = 0$. In other terms, no processing of $\mathbf{Y}$, deterministic or random, can increase the information that $\mathbf{Y}$ contains about $\mathbf{X}$.*

# B    Training Dynamics

As explained in Section 3.1, we train each perturbation function $T_{\mathbf{z}}^i$ separately, and initialize as the identity function. This allows InfoNCE loss in Equation (5) to start from a very low value. During training, $T_{\mathbf{z}}^i$ progressively enhances diversity in the views, reducing $I(\mathbf{X}; \mathbf{X}')$, and causing InfoNCE to increase over time. An example of these training dynamics is illustrated in Figure 6a. Figure 6b shows some qualitative examples of how views in the pixel space evolve during training. Specifically, the image on the left represents the anchor, while other images show the corresponding perturbed view as the magnitude increases.

For identity initialization, we follow a Gaussian distribution $\mathcal{N}(0.0, 0.01)$ for the weights and a Uniform distribution $\mathcal{U}(-0.001, 0.001)$ for the biases, implying $T_{\mathbf{z}}^i(\mathbf{z}) \approx \mathbf{z}$. We use a batch size of 64, Adam (Kingma & Ba, 2015) optimizer with $\beta_1 = 0.5, \beta_2 = 0.999$ and a temperature $\tau = 0.1$ in the InfoNCE loss (Oord et al., 2018).

Regarding training time, each $T_{\mathbf{z}}^i$, a two-layer MLP, requires approximately two hours of training on one A-100 GPU. At inference time, we measure that Monte Carlo sampling of 100K latents takes no more than 100 seconds per latent level, using an RTX 2080 TI GPU.

# C    Continuous Sampling

We detail the batch generation procedure for anchors and positive views in Algorithm 1. As explained in the main paper, the algorithm requires loading the pre-trained generative model (MLVGM in our case) on the same GPU device as the encoder model to be trained. In this way, the generated batch will be already on the correct device, eliminating the need for data loading.

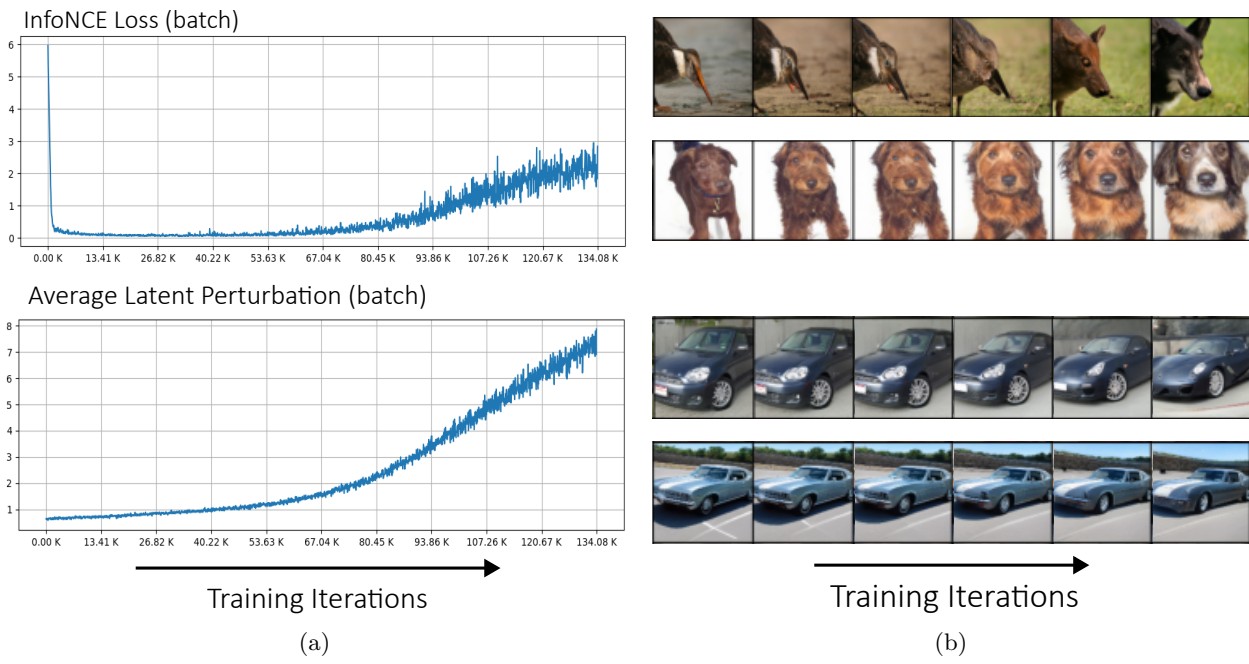

(a)

(b)

Figure 6: **(a)** Example of training dynamics comparing the InfoNCE Loss (top) and the average latent perturbation (bottom) for a single batch. $X$ axis denotes training iterations. **(b)**. Qualitative examples of the generated views' evolution during training (anchor view on the left).

---

**Algorithm 1** Continuous Sampling of Batch (training time)

---

**Input**: MLVGM model $g$ with $n$ latent variables; anchors sampling function $N^t(\mu, \sigma, t)$; positive perturbation functions (*random* or *learned*) $T_{\mathbf{z}}^i$.
**Parameter**: batch size $B$.
**Output**: $\mathcal{A}, \mathcal{P}$ Generated batch of anchors and positives.

1:   $\mathcal{A} \leftarrow \emptyset$                                                                ▷ initialize empty anchors set
2:   $\mathcal{P} \leftarrow \emptyset$   ▷ initialize empty positives set
3:   **for** $b \in \{0, \ldots, B-1\}$ **do**
4:       $A_b \leftarrow \emptyset$   ▷ initialize empty set of anchor variables
5:       $P_b \leftarrow \emptyset$   ▷ initialize empty set of positive variables
6:       **for** $i \in \{0, \ldots, n-1\}$ **do**
7:          $a_i \sim N^t(\mu, \sigma, t)$   ▷ sample anchor code
8:          $p_i \leftarrow T_{\mathbf{z}}^i(a_i)$   ▷ generate positive code
9:          $A_b = A_b \cup \{a_i\}$
10:         $P_b = P_b \cup \{p_i\}$
11:       **end for**
12:       $\mathbf{a}_b \leftarrow g(A_b)$   ▷ generate anchor image
13:       $\mathbf{p}_b \leftarrow g(P_b)$   ▷ generate positive image
14:       $\mathcal{A} = \mathcal{A} \cup \{\mathbf{a}_b\}$
15:       $\mathcal{P} = \mathcal{P} \cup \{\mathbf{p}_b\}$
16:   **end for**
17:   **return** $\mathcal{A}, \mathcal{P}$   ▷ SSCRL Encoder's inputs

---

# D View Generation Training Details

The code for this work has been developed using two popular `python`'s deep learning libraries: `pytorch` (Paszke et al., 2019) and `pytorch lightning` (Falcon & The PyTorch Lightning team, 2019). The BigBiGan generator's code and weights have been introduced in (Melas-Kyriazi et al., 2022), and can be obtained at (Melas-Kyriazi, 2021). For StyleGan2, the official `github` repositories are available, specifically for code (Karras & Hellsten, 2021) and weights (Karras & Hellsten, 2019). In the following, we report the training-specific details used in the implementation.

**Training and evaluating encoders.** We train each SSCRL encoder (regardless of the framework) for 100 epochs using SGD optimizer with momentum 0.9 and weight decay $1 \times 10^{-4}$. The learning rate is set as $0.05 \times \text{BatchSize}/256$, with a cosine decay scheduler and an additional linear warmup for the first 10 epochs if $\text{BatchSize} \geq 1024$. Specifically, the BigBiGan encoders are trained with a batch size of 1024, while for StyleGan2 we use a batch size of 512, due to the generator's larger number of parameters. Matching the output resolution of BigBiGan and following prior studies (Jahanian et al., 2021; Li et al., 2022), all views have $128 \times 128$ image resolution.

For sampling positives, the *random* baselines are trained with $\mathcal{N}^t(0.0, 0.2, 2.0)$ and $\mathcal{N}^t(0.0, 0.25, 0.9)$ for each generator, respectively BigBiGan and StyleGan2. Following the suggestions of the original work (Li et al., 2022), we monitor the generated views and select the best checkpoints for the *learned* baselines. For our ML views, BigBiGan samples positives by fixing the first latent and with parameters $\mathcal{N}^t(0.0, 1.0, 2.0)$ for the remaining codes (including for simplicity the unused latent). On the other hand, we perturb the StyleGan2 selected groups with parameters: $\mathcal{N}^t(0.0, 0.2, 1.0)$; $\mathcal{N}^t(0.0, 0.1, 1.0)$; $\mathcal{N}^t(0.0, 0.8, 1.0)$; $\mathcal{N}^t(0.0, 0.8, 1.0)$, respectively. Similar to baselines, the *learned* perturbations are decided by monitoring checkpoints. In general, they follow the same perturbation magnitudes as the *random* counterpart.

All linear classifiers used for BigBiGan evaluation are trained for 60 epochs with a batch size of 256, SGD optimizer, and a learning rate of 30.0 with cosine decay. StyleGan classifiers follow the same setup, but they are trained for 100 epochs. For Pascal VOC detection, the R50-C4 backbone is fine-tuned for 24000 iterations on `trainval07+12` split and evaluated on `test07`.

**Data and preprocessing.** All our experiments use `FFCV` (Leclerc et al., 2023) library for efficient data storage and fast loading. ImageNet-1K's images are stored at $256 \times 256$ resolution and resized to $128 \times 128$ during loading, to match the output resolution of BigBiGan. Regarding LSUN Cars/StyleGan2, instructions to download the 893K training images can be found at (Karras & Hellsten, 2021). These are $512 \times 384$ resolution images, which are stored at $512 \times 512$ with black padding, to match StyleGan2 outputs. During loading, images are first center cropped at $384 \times 384$, removing padding, and then resized at $128 \times 128$. The same preprocessing is applied to the generated images.

The data augmentation and preprocessing pipelines rely on the `kornia` library (Riba et al., 2020). During transfer classification learning, we apply random resize crop and random horizontal flip during training, and center crop during validation/testing. In all experiments, images are normalized with ImageNet mean and standard deviation values, and the final size (after cropping) is $112 \times 112$.

**Hardware resources and reproducibility.** Most experiments have been run using 4 NVIDIA A100-SXM4-40GB GPUs, with an exception for the StyleGan2-trained encoders, which required 8 GPUs of the same type, due to the larger number of parameters of StyleGan2 w.r.t. BigBiGan. Other minor experiments, like the training of perturbation functions, required only 1 GPU. To ensure reproducibility, random seeds have always been fixed. For Continuous Sampling, seeds are changed at every iteration and device-specific. This procedure avoids reproducing the same batches during training but allows for consistency throughout different runs.

**Perturbations in $\mathcal{Z}$ and $\mathcal{W}$ space.** In BigBiGan experiments, perturbations in the latent space are applied by summing the noise vector to the selected latents, as described in Section 3 of the main paper. For StyleGan2, the final latent space is known as $\mathcal{W}$, and a mapping network is used to perform $f(\mathbf{z}) = \mathbf{w}$.

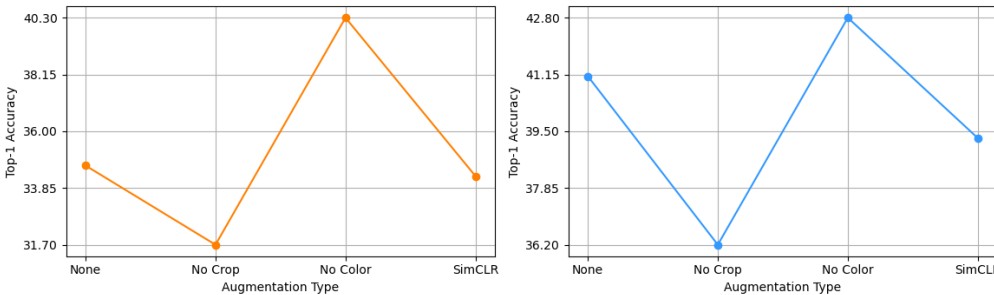

Figure 7: Ablation study on the $T_{\mathbf{x}}$ pixel-space augmentations combined with BigBiGan's generated views. Left (orange) represents *random* generated views while right (blue) represents *learned* generated views.

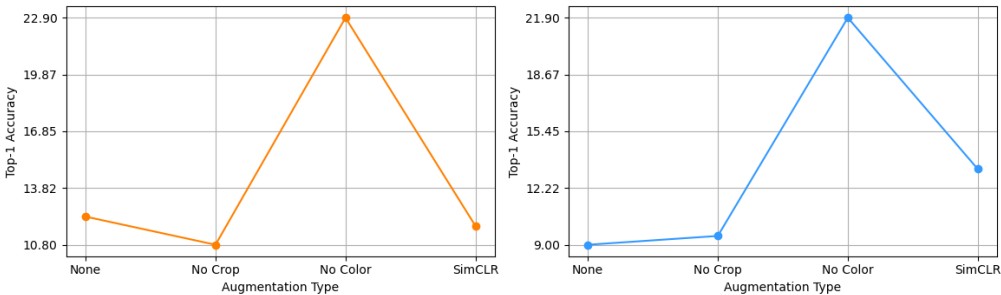

Figure 8: Ablation study on the $T_{\mathbf{x}}$ pixel-space augmentations combined with StyleGan2's generated views. Left (orange) represents *random* generated views while right (blue) represents *learned* generated views.

Here, $f$ is the mapping network, $\mathbf{w}$ the random latent vector in the $\mathcal{W}$ space, and $\mathbf{z}$ the initial random vector sampled from a truncated Gaussian distribution ($\mathcal{Z}$ space). The positive views in this case are obtained as $f(T_{\mathbf{z}}(\mathbf{z})) = \mathbf{w}$, where $T_{\mathbf{z}}$ is a *random* or *learned* perturbation that affects only the selected variables.

## E  Ablation Studies

Previous studies (Jahanian et al., 2021; Li et al., 2022) show that the *random* and *learned* baselines still benefit from the application of $T_{\mathbf{x}}$ SimCLR augmentations on top of the generated views. In detail, they perform some ablation studies training encoders on Imagenet-100 and a subset of LSUN cars, where different combinations of pixel-space augmentations are tested: cropping and horizontal flipping, grayscale and color jittering, none of the previous or all of them. When utilizing ML perturbations, we qualitatively observe that the generated views can assume various realistic colors (as seen for example in Figure 4 of the main paper). For this reason, in the experimental analysis, we only use cropping and flipping as pixel space augmentations.

To validate our hypothesis, in Figures 7 and 8 we report the result of the same experiments of previous work, but performed on our ML views. In all cases, removing color jittering and grayscale operations increases the final linear evaluation Top-1 accuracy. These observations empirically confirm that ML views do not need additional color transformations.

## F  NVAE Experiments

In the main paper, experiments are conducted on two state-of-the-art GAN methods. To showcase the general properties of the proposed approach, we conducted an additional ablation study where we trained a small NVAE (Vahdat & Kautz, 2020) with 24 latent variables, organized in 3 groups, on Cifar10 (Krizhevsky et al., 2009) ($32 \times 32$ resolution). The model achieves a FID of 23.63, and an IS score of $6.36 \pm 0.16$.

Table 4: Results of the MC simulation on the NVAE model pre-trained on Cifar-10. For each latent group $g$ we show the final InfoNCE loss value and the estimated mean ($\mu$) and standard deviation ($\sigma$) of the corresponding inferred distribution.

| latent group | loss | estimated $q^i$ | |
|:---:|:---:|:---:|:---:|
| $(g)$ | (InfoNCE) | $(\mu)$ | $(\sigma)$ |
| 1 | 0.60 | 0.04 | $1e-3$ |
| 2 | 0.60 | 0.96 | 0.41 |
| 3 | 0.60 | 1.18 | 1.25 |

Table 5: Evaluation on Cifar10 (Krizhevsky et al., 2009) of encoders trained using an NVAE (Vahdat & Kautz, 2020) with 3 latent groups as a data source.

| Encoder | $T_{\mathbf{z}}$ | Top-1 Accuracy (%) |
|:---|:---|:---:|
| Baseline real | - | 79.15 |
| Baseline synth | - | 54.95 |
| Random | $N^t(0., 0.2, 1.)$ | 59.44 |
| Random | $N^t(0., 0.3, 1.)$ | 59.61 |
| Random | $N^t(0., 0.4, 1.)$ | 59.34 |
| ML Random | $N^t(0., 0.3, 1.)$ | 59.21 |
| ML Random | $N^t(0., 0.4, 1.)$ | 60.75 |
| ML Random | $N^t(0., 0.5, 1.)$ | 59.98 |

At first, we measure the impact of each latent group replicating the experiments of Table 1. The results, included in Table 4, confirm that the "global-to-local" trend is valid also in this case. Moreover, we observe that the first group is highly influencing the final outcome with respect to the remaining groups.

Afterwards, we test the model's abilities as a data source, replicating the experiments of Table 2 for the *random* case with SimSiam framework and Continuous Sampling. Table 5 shows the final Top-1 and Top-5 accuracies obtained on the Cifar10 evaluation set. In this case, the absolute best result is achieved by the same encoder trained on real data (79.15% accuracy), with a large gap (54.95%) when using NVAE as a data source without further latent perturbations. Adding the same perturbations to all groups provides an immediate accuracy boost (+5%), with various standard deviations. After observing the results in Table 4, we decided to fix the first group (overused), and find that applying larger perturbations to groups 2-3 only allows for a larger std and additional boost, with a final accuracy of 60.75%. These experiments prove the feasibility of our methods also on Autoencoder-based MLVGMs, despite the small model employed.

# G   Transfer Learning

Table 6 contains the results for each transfer classification learning experiment performed on top of the SimSiam pre-trained encoders using BigBiGan / ImageNet-1K. The results refer to the last column of Table 2, where only the mean Top-1 accuracy over the 7 target datasets has been reported. For a better comparison of encoders' generalization capabilities, each run was tested with 5 different seeds, and the mean Top-1 accuracy was taken.

On each dataset, the results are computed using the test set where available, otherwise on the validation set, maintaining the original splits. For DTD (Cimpoi et al., 2014) the first split between the proposed ones has been employed, while for Caltech101 (Fei-Fei et al., 2004) we selected a random split of 30 train images per class, using the remaining for testing. In this case, all the background images (distractors) have been removed.

| Encoder | Top-1 Accuracy on Target Dataset | | | | | | |
|---|---|---|---|---|---|---|---|
| | Birdsnap | Caltech101 | Cifar100 | DTD | Flowers102 | Food101 | Pets |
| Baseline real | $\mathbf{63.1 \pm 0.3}$ | $\mathbf{83.1 \pm 1.0}$ | $26.2 \pm 0.7$ | $\mathbf{\underline{56.4 \pm 0.3}}$ | $\mathbf{59.8 \pm 2.6}$ | $\mathbf{\underline{51.5 \pm 0.2}}$ | $\mathbf{67.6 \pm 0.4}$ |
| Baseline synth | $46.3 \pm 0.4$ | $67.9 \pm 1.6$ | $\mathbf{33.4 \pm 0.4}$ | $47.7 \pm 0.5$ | $46.7 \pm 0.9$ | $41.6 \pm 0.2$ | $47.1 \pm 1.2$ |
| random | $45.4 \pm 0.3$ | $68.9 \pm 0.9$ | $31.7 \pm 0.7$ | $47.9 \pm 0.6$ | $46.4 \pm 0.8$ | $42.3 \pm 0.3$ | $46.5 \pm 1.3$ |
| ML random | $\mathbf{\underline{64.3 \pm 0.6}}$ | $\mathbf{\underline{84.4 \pm 0.3}}$ | $\mathbf{\underline{41.1 \pm 0.8}}$ | $\mathbf{54.9 \pm 1.0}$ | $\mathbf{\underline{63.2 \pm 0.6}}$ | $\mathbf{50.1 \pm 0.3}$ | $\mathbf{59.5 \pm 0.3}$ |
| learned | $42.0 \pm 0.3$ | $72.9 \pm 0.8$ | $31.7 \pm 0.8$ | $48.8 \pm 0.5$ | $45.2 \pm 0.7$ | $38.6 \pm 0.4$ | $44.0 \pm 0.8$ |
| ML learned | $\mathbf{57.7 \pm 0.5}$ | $\mathbf{77.5 \pm 0.5}$ | $\mathbf{36.7 \pm 0.4}$ | $\mathbf{53.1 \pm 0.4}$ | $\mathbf{60.3 \pm 0.6}$ | $\mathbf{47.6 \pm 0.5}$ | $\mathbf{51.8 \pm 0.2}$ |

Table 6: Transfer classification Top-1 accuracy's results for each pre-trained encoder using the SimSiam framework and BigBiGan/ImageNet-1K as a data source. 7 different target datasets have been tested.

# H  Qualitative Visualizations

Figure 9 and Figure 10 show some examples of generated views using BigBiGan and StyleGan2, respectively. In each Figure, rows represent different views, and columns display (from left to right): the initial anchor image, *random vs. learned* baselines, and the views obtained with our method. All the examples are generated using the same hyperparameters, which were also used for training the contrastive encoders. In order to allow a better comparison of the different methods, no further $T_{\mathbf{x}}$ augmentations are applied on top of the generated images. As depicted in both figures, our generated views can produce a wide range of transformations, including realistic color changes that allow us to eliminate the color jittering $T_{\mathbf{x}}$ augmentation.

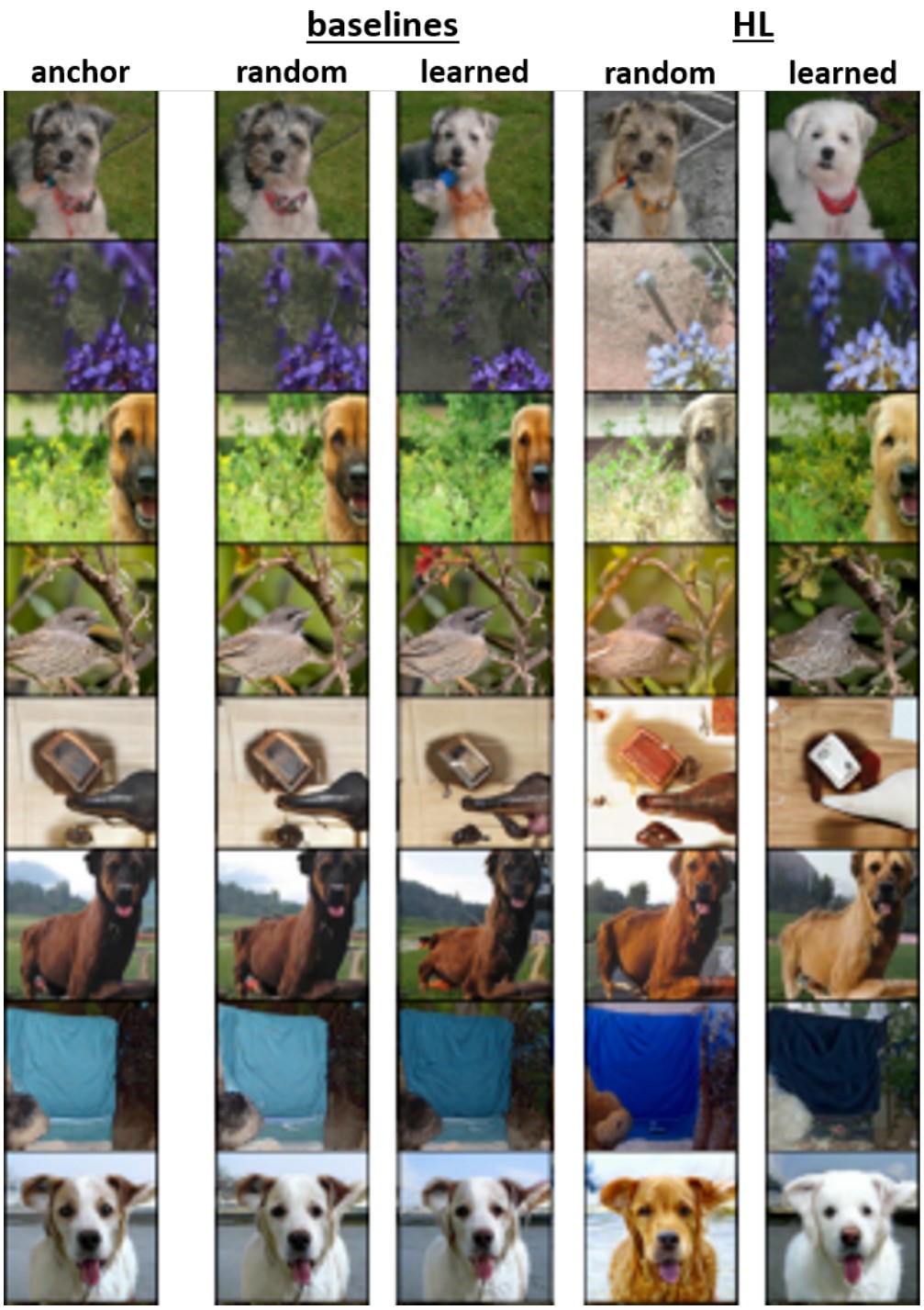

Figure 9: Example of generated views from BigBiGan anchors. From left to right: anchor, *random* and *learned* baseline views, *random* and *learned* ML views.

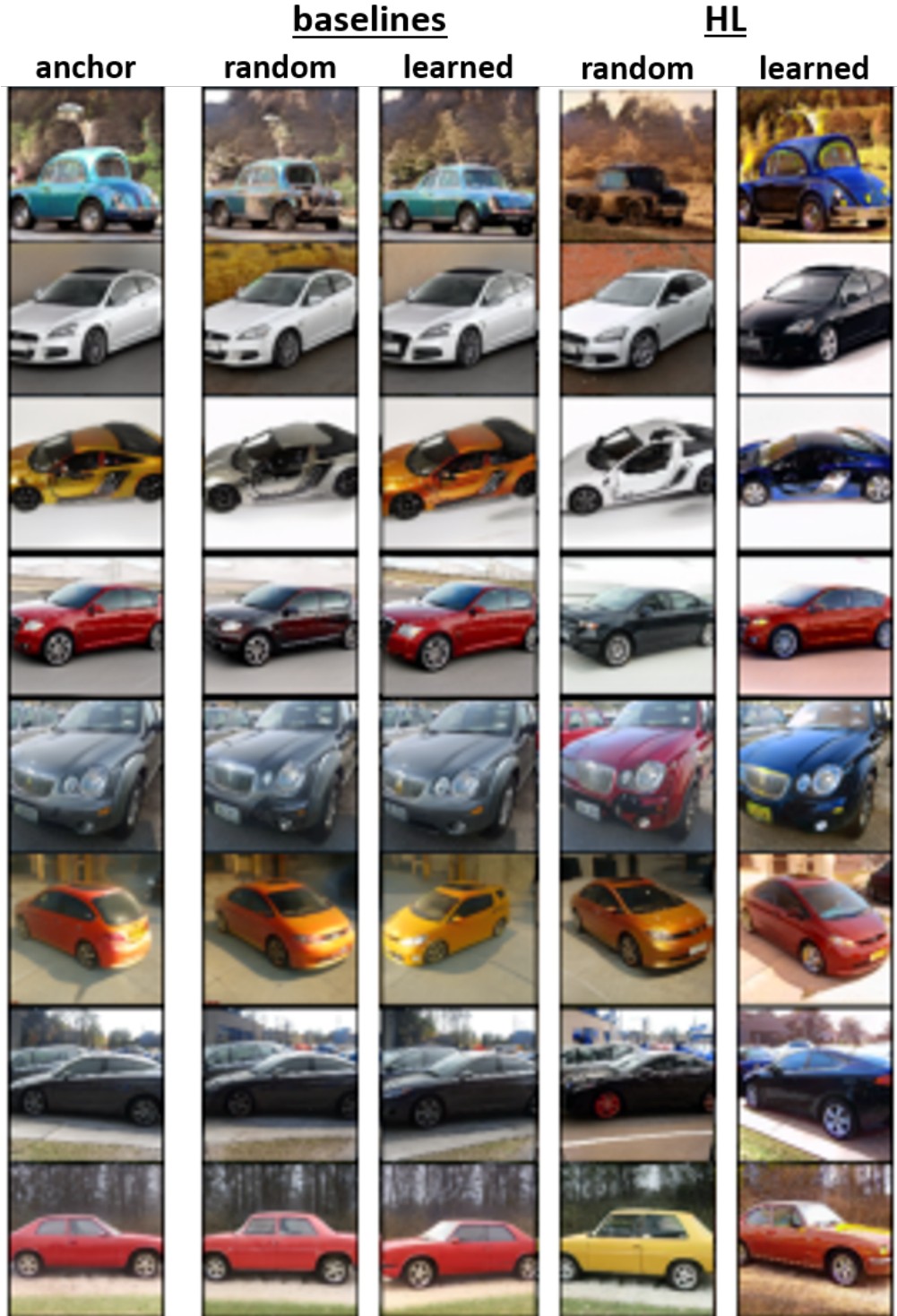

Figure 10: Example of generated views from StyleGan2 anchors. From left to right: anchor, *random* and *learned* baseline views, *random* and *learned* ML views.