# OpenReview forum: "A Mutual Information Perspective on Multiple Latent Variable Generative Models for Positive View Generation"
_TMLR — Accepted by TMLR_

### Review · Reviewer_KaX2 · 2025-02-27

**Summary Of Contributions:**

1) the authors introduce a contrastive loss based method for quantitatively evaluating the impact of the different latent varibale perturbations in Multi Latent Cariable Generative Models MLVGMs.
2) they use MLV-GMs to generate positive samples for self-supervised contrastive representation learning.
3) they generate these training samples in real time instead of saving them to disk before training their Self-Supervised Contrastive Representation Learning SSCRL models.

**Audience:**

Yes

**Claims And Evidence:**

Yes

**Requested Changes:**

All of these suggestions are made with low confidence; follow them if they seem right to you:

MLVGM and SSCRL are some really rough acronyms. My brain regards them like chaotic piles of capital letters. Maybe a hyphen would help, MLV-GMs?

Definition 1: Not sure I agree with this use of the word "random", IIUC, at least for stylegan, the latent z's {z1,...zn} are themselves pulled from a learned joint distribution and probably have high mutual information. Is it fair to call them "random"?

An odd detail about their setup: if T(z) is a deterministic function, and assuming g(z1,...zn) is invertible, then the true ideal InfoNCE loss between I1 = f(g(z1,...zi,...zn)) and I2 = f(g(z1,...T(zi),...zn)) should always be... zero? Since there exists a deterministic function from I1 to I2. Any loss greater than zero just means that your encoder f has failed to learn this function. Would be more principled if T(z) included some random noise as input?

**Strengths And Weaknesses:**

I had to read almost the entire paper to understand what it was really about. And I'm still not sure I entirely understand what exactly the first experiment about mutual information proves or enables. The abstract and introduction could do a better job of clearly explaining what the authors did, why they did it, and what conclusions they drew.

Their first method and experiment, about training latent perturbation models to achieve similar contrastive losses across layers, feels very complicated compared to the insights it cashes out in. AFAICT, the takeaway from these experiments is effectively that you need to apply much larger perturbations to the later latent variables than the earlier ones in order to achieve the same level of deviation from the original image. It feels like qualitatively similar conclusions could have been reached using far simpler experiments. For example, you could just plot the LPIPS between the original and perturbed images as a function of the magnitude of a random perturbation to each latent. I feel like this very simple experiment would probably have yielded the same final insight as this complicated adversarial contrastive learning training procedure.

A few times in the abstract and introduction, the authors imply that the results of their first experiment about mutual information informed the way they trained their SSRCL in the second experiment. I am not convinced that the mutual information-based analysis of MLVGMs is actually informing your SSCRL method for generating positive views. Indeed the conclusion from your discussion in section 3.2 appears to be that this analysis is ultimately inadequate for determining what perturbations to use. So what was the utility of that mutual information analysis described in section 3.1? It seems like the final takeaway was mainly how much to perturb each latent, but it's not clear precisely *how* the results from table 1 informed those hyperparameter choices from appendix D.

Tables 2 and 3 do convincingly demonstrate that the author's MLVGM-based positive view generation usually outperforms their baseline based on more standard augmentations.

Not convinced that Continuous Sampling counts as a technically novel contribution to the field. Feels more like an implementation detail of their training setup.

---

> ### Author Response · Authors · 2025-03-24
>
> Dear Reviewer KaX2,
>
> Thank you for taking the time to read our work and for providing constructive feedback to help improve it. We greatly appreciate your insights.
>
> We will provide a detailed response to each specific concern once all reviews are available. In the meantime, we are actively working on improving certain parts of the paper,
> including Section 3.2 and others.
>
> In particular, following your suggestion regarding the clarity of the first experiments’ purpose, we are actively working on enhancing its presentation.
> However, we would appreciate further clarification on which specific aspects of the abstract and introduction you find unclear.
>
> For instance, in the initial part of the abstract, we state what we did: _"we propose a novel framework to
> systematically quantify the impact of each latent variable in MLVGMs, using Mutual Information (MI) as a
> guiding metric"_; why we did it: _"Yet their (of MLVGMs) generative dynamics and latent variable utilization remain
> only empirically observed"_; and our conclusions: _"Our analysis reveals underutilized variables and
> can guide the use of MLVGMs in downstream applications."_
>
> To ensure we address your concerns effectively, a more specific indication of which elements require further
> clarification would be extremely helpful.
>
> Thank you again for your thoughtful feedback.

---

> ### Author Response · Authors · 2025-07-05
>
> Dear Reviewer,
>
> We did our best to enhance the clarity of our paper in the revised version, as we understand that some parts may be challenging for some readers. In particular, we included revisions in abstract, introduction and methodologies.  We hope that the new version will result easier to read and understand.
>
> In short, we present a systematic procedure to quantify the impact of individual latent variables within Multiple Latent Variable Generative Models (MLVGMs) while using InfoNCE as a proxy to measure MI shifts between images. We show that the hierarchical disentanglement within MLVGMs can be leveraged to generate positive views for Self-Supervised Contrastive Representation Learning (SSCRL). Moreover, we introduce a continuous sampling strategy which dynamically generates synthetic data during training, aiming to enhance data diversity.
>
> The remaining concerns are addressed in the following.
>
> **Using LPIPS instead of InfoNCE**
>
> The main difference between using LPIPS and Mutual Information relies in their meaning. Mutual Information has a precise mathematical/probabilistic significance. That is, it measures the reduction in uncertainty about one random variable when another is known. LPIPS instead, is a perceptual metric that is believed to well-align with human judgment. However, it has no mathematical formulation or theoretical limitations, which would significantly reduce the value of our experiments, making them more qualitative than quantitative.
>
> In more practical terms, while it is guaranteed that MI will shift for any non-deterministic function, with LPIPS several problems may arise:
>
> I) Standard LPIPS employs VGG-16 pre-trained on ImageNet as a pre-trained feature extractor. How precise would this be for different domains (e.g. LSUN Cars in our case)?
>
> II) How does LPIPS behave in different transformations, like color changes, translations, rotations, zoom in/out?
>
> Experimentally, we tried to measure LPIPS for the six latent levels of BigBiGAN on different magnitudes. To match the result of MI, we expect LPIPS to decrease between latents on each row. This is not the case between latents 1 and 2, suggesting that LPIPS considers more important color changes (operated by latent 2), than shape modifications (operated by latent 1). The result is even more evident for larger perturbations, where LPIPS for latent 3 (color changes) is the highest.
>
> | **Perturbation Magnitude** | **Latent 0** | **Latent 1** | **Latent 2** | **Latent 3** | **Latent 4** | **Latent 5** |
> |----------------------------|--------------|--------------|--------------|--------------|--------------|--------------|
> | 0.5                        | 0.1484       | 0.0083       | 0.0091       | 0.0005       | 0.0001       | 0.0000       |
> | 1.0                        | 0.2691       | 0.0247       | 0.0283       | 0.0018       | 0.0002       | 0.0000       |
> | 2.0                        | 0.4190       | 0.0635       | 0.0764       | 0.0063       | 0.0009       | 0.0000       |
> | 4.0                        | 0.5663       | 0.1389       | 0.1698       | 0.0207       | 0.0032       | 0.0000       |
> | 8.0                        | 0.6792       | 0.2582       | 0.3137       | 0.0604       | 0.0114       | 0.0000       |
> | 16.0                       | 0.7393       | 0.4084       | 0.4834       | 0.1543       | 0.0366       | 0.0001       |
> | 32.0                       | 0.7644       | 0.5682       | 0.6347       | 0.3323       | 0.1035       | 0.0002       |
> | 64.0                       | 0.7803       | 0.6839       | 0.7484       | 0.5720       | 0.2511       | 0.0009       |
> | 128.0                      | 0.7948       | 0.7402       | 0.8047       | 0.7701       | 0.4814       | 0.0033       |
> | 256.0                      | 0.8048       | 0.7720       | 0.8196       | 0.8435       | 0.6930       | 0.0119       |
>
> **How the first contributions informs the decision for views**
>
> We revised Section 3.2 of the paper to clarify the role of MI analysis in the selection of perturbation magnitudes. In particular, it is employed as a preemptive step to discard under or over utilized codes. This allows us to discard non-informative latents from the decision process.
>
> **Continuous Sampling**
>
> While Continuous Sampling requires several implementations choices, well discussed in the paper, the value of its contribution arises from the motivations for its introduction. In fact, with CS we empirically demonstrated that it is not necessary to pre-generate large quantities of synthetic data before training, differently from what previous work has always proposed.

---

> ### Author Response · Authors · 2025-07-05
> **Addressing minor concerns**
>
> **Random in Definition 1**
>
> In this context, the term "random" refers to the fact that each $\mathbf{Z}_i$ denotes a "random variable" in the latent space, in the probabilistic sense. However, it does not imply that each $\mathbf{Z}_i$ is independent from the others.
>
> **InfoNCE should be 0**
>
>  While $T(\mathbf{z})$ is deterministic, it is not obvious to assume that the generator represents an invertible function. On the opposite, there is no formal proof that latent variable generative models are invertible, and only approximations can be learned by joint training, as in [1].
>
>  Alternatively, one can reason on the fact that InfoNCE is computed on the generated views in the pixel space, which typically differ in shape, color, background, position etc... All these transformations are typically non invertible, and therefore it is expected to have InfoNCE > 0.
>
>  [1] Ghosh, Partha, et al. "Invgan: invertible gans." DAGM German Conference on Pattern Recognition. Cham: Springer International Publishing, 2022.

---

> > ### Comment · Reviewer_KaX2 · 2025-08-05
> > **Response to authors**
> >
> > Thank you for performing these LPIPS experiments. It’s surprising to see that the relationship between latents 1 and 2 are inverted compared to mutual information.
> >
> > And thanks for the expansion of section 3.2. I understand your motivation for choosing this mutual information objective significantly better now. It’s inspired by the relevance of mutual information in the contrastive learning regime.
> >
> > Re: InfoNCE=0
> >
> > I have thought about this more and had some insights about your infoNCE loss.
> >
> > Upon reviewing this argument, I realize that you would also need to be able to recover z_i given T(z_i) for this argument to hold. But it sounds like you agree that *if* these assumptions held, then your InfoNCE loss should always be zero?
> >
> > If these functions are invertible, the ideal feature extractor behavior will effectively be to invert the generator and map T(z_i) and z_i to the same vector. Like basically you’re trying to recover z1,...zn from both g(z1,...zn) and g(z1,...T(zi),...zn). The representation you are learning is therefore very close to the GAN’s latent representation of your image, with some information about the perturbed latent being lost.
> >
> > This understanding suggests another body of related work: using GAN inversion as an image feature extractor. For example, the BigBiGan paper, which you use, uses their image encoder (i.e. GAN inverter) to extract features for linear ImageNet classification. I think their numbers may be slightly better but I haven’t compared deeply enough to know if it’s apples-to-apples. But I strongly suspect that your feature extractor is going to learn very similar features to the BigBiGan inverter, but with the perturbed latent sort of “blurred”.
> >
> > If I am correct, then any nonzero loss in your objective function comes from one of two sources: 1) the loss of information associated with applying g() and T(), or 2) the failure of f() to accurately invert these functions.

---

> > > ### Author Response · Authors · 2025-08-12
> > >
> > > Dear Reviewer,
> > >
> > > We are glad you appreciated the additional experiment and the new version of the paper. We believe these changes have improved the quality of our work.
> > >
> > > We are also grateful for your feedback and for engaging in the discussion on InfoNCE loss. Upon a very interesting internal debate, we arrived at the considerations expressed below.
> > >
> > > **you agree that if these assumptions held, then your InfoNCE loss should always be zero?**
> > > You are right: actually, in the hypothesis that both $T()$ and $g()$ are fully invertible, the real mutual information should be maximized, and therefore InfoNCE = 0. This would correspond to a scenario where $z$ and $x$ are two information-equivalent representations of the same image.
> > >
> > > **If these functions are invertible, the ideal feature extractor behavior will effectively be to invert the generator...**
> > > In this hypothesis, the generated views $g(z1,...zn)$ and $g(z1,...T(zi),...zn)$ have maximal mutual information. Hence, the ideal feature extractor $f()$ would reach a representation $e$ that is also information-equivalent to $x$ and $z$, and $f()$ can be considered an inverter function.
> > >
> > > **The representation you are learning is therefore very close to the GAN’s latent representation of your image, with some information about the perturbed latent being lost.**
> > > If $T()$ and $g()$ are invertible, the learned representation should be exactly the same as the GAN's latent one, since applying the perturbation $T()$ does not lose any information.
> > >
> > > **Using GAN inversion as an image feature extractor**
> > > This is an interesting connection, which has been already explored in [1] (reported in our paper as the "random" method).  In Section 4.2 they explicitly state: "*Note the BigBiGAN encoder is jointly trained with the generator, thus requiring access to the original real dataset during training. This is **not** the setting we are targeting*".
> > >
> > > Since our setting is the same as [1] (we do not use real data), comparing directly with BigBiGan encoder is not exactly "apples-to-apples".
> > >
> > > However, in [1] they train an inverter function on generated data (an "apples-to-apples" scenario), reaching a top-1 accuracy of 26.43% (see their Table 1), which is far below our 53.7% (see our Table 2). This shows that the representations learned by our method are not equivalent to those of an inverter on g().
> > >
> > > [1] Ali Jahanian, Xavier Puig, Yonglong Tian, and Phillip Isola. Generative models as a data source for multiview
> > > representation learning. In International Conference on Learning Representations, 2021.
> > >
> > > **I strongly suspect that your feature extractor is going to learn very similar features to the BigBiGan inverter**
> > > As previously stated, only in the hypothesis that everything is invertible, then, all the representations are equivalent. However, the previous result shows that it is not our case.
> > >
> > > **any nonzero loss in your objective function comes from one of two sources: 1) the loss of information associated with applying $g()$ and $T()$, or 2) the failure of $f()$ to accurately invert these functions.**
> > > In the hypothesis that both $g()$ and $T()$ are invertible, then any nonzero loss is caused by 2) the failure of learning a proper $f()$.
> > >
> > > However, as we have shown in our previous discussion, since $g()$ is not invertible, then the nonzero loss is caused by 1) in our scenario.

---

### Review · Reviewer_Smc8 · 2025-03-20

**Summary Of Contributions:**

This paper explores how mutual information is distributed across latent spaces in multiple latent variable generative models (MLVGMs). The authors propose a principled approach to quantify the “global-to-local” information bottleneck by measuring mutual information (MI) after perturbations at different levels. To achieve this, they introduce an adversarial method that learns perturbations at each hierarchical level, ensuring a constant deviation in MI across levels. Additionally, the paper extends this analysis to the domain of contrastive learning, where the authors propose using their framework to generate positive views for contrastive representation learning.

**Audience:**

Yes

**Claims And Evidence:**

Yes

**Requested Changes:**

Based on the aforementioned weaknesses, I suggest the following revisions, ordered by priority:
- Improve clarity, structure, and flow. Each section should focus on addressing a specific aspect of the paper. Identifying and removing redundant text would enhance readability and make the contributions clearer.
- Clarify the connection between the information-theoretic framework for learning balanced perturbations and the view generation approach for contrastive learning. This relationship should be presented in a more structured and principled manner to improve coherence.
- Evaluate the method on Hierarchical VAEs, such as NVAE or VDVAE (missing reference). This is particularly relevant since the proposed method could benefit from fine-tuning a pre-trained encoder in these architectures.

Reference:

[1] Child, Rewon. “Very Deep VAEs Generalize Autoregressive Models and Can Outperform Them on Images.” International Conference on Learning Representations (ICLR).

**Strengths And Weaknesses:**

### Strengths
- The paper presents a novel and principled approach to quantifying the flow of MI across hierarchical latent spaces in MLVGMs.
- The adversarial perturbation method provides a structured way to analyze information bottlenecking at different levels, offering a deeper understanding of global-to-local MI compression.
- The proposed framework is well-motivated and theoretically sound, with a clear focus on quantifying MI variations in hierarchical generative models.
- The paper extends its analysis to contrastive learning, proposing a new approach for generating positive views to improve representation learning.
- The experimental evaluation demonstrates the feasibility of the method, primarily on GAN-based architectures.

### Weaknesses
- The connection between the MI analysis and its application to contrastive learning is unclear. The contrastive learning contribution appears disconnected from the MI perturbation framework, and it is not immediately evident how the first method informs the second.
- The writing quality and organization need significant improvement. The paper is highly redundant, repeatedly cycling through the same claims, which disrupts readability and makes it difficult to pinpoint the main contributions within each section. A major revision to improve clarity and flow and delete redundant text is recommended.
- The paper presents MLVGMs as a general class of models encompassing GANs and VAEs, yet the empirical evaluation is conducted only on GAN-based architectures. This limits the scope of the claims regarding the applicability of the proposed method to MLVGMs in general.

---

> ### Author Response · Authors · 2025-03-24
>
> Dear Reviewer Smc8,
>
> Thank you for taking the time to read our work and for providing constructive feedback. We greatly appreciate your insights.
>
> We are actively working on incorporating experiments with VAE-based methods, particularly NVAE, and improving various sections of the paper, including Section 3.2,
> to better clarify the connections between our first and second contributions.
>
> Regarding your suggestion on improving the overall quality of the paper, we acknowledge that some parts may be redundant, especially for readers well-versed in the field. However, we would greatly appreciate more specific feedback in this regard. In particular, which sections do you believe require the most urgent revisions?
>
> Thank you again for your valuable suggestions.

---

> ### Author Response · Authors · 2025-07-05
>
> Dear reviewer,
>
> Given your feedback, we revised some parts of the paper, with explicit focus on clarity and redundancies removal. We also tried to better clarify contributions within each section. In particular, we greatly revised Section 3.2, stressing the connection between the MI analysis and its application to contrastive learning. In short, the MI analysis immediately reveals the presence of over or under utilizes codes, allowing us to discard non-informative latents from the process.
>
> Regarding experiments on VAE-based MLVGMs, our original submission included an ablation study on NVAE in Appendix F. However, we extended the study in the revised version of the manuscript. In detail, we measured the impact of each latent group (3 in total), showing that the "global-to-local" behavior is consistent also in this scenario. Moreover, we show that generating positive views by fixing the first latent group is beneficial, and allows for larger perturbations to the remaining groups.

---

### Review · Reviewer_jyeL · 2025-06-30

**Summary Of Contributions:**

This paper presents a systematic procedure to quantify the impact of individual latent variables within Multiple Latent Variable Generative Models (MLVGMs) while using InfoNCE (at embeddings level) as a proxy to measure MI shifts between images. Building on this, while leveraging hierarchical disentanglement within MLVGMs, the authors introduce an approach to generate positive views for Self-Supervised Contrastive Representation Learning (SSCRL). They further introduce a continuous sampling strategy which dynamically generates synthetic data during training, aiming to enhance data diversity. Experiments on BigBiGAN, StyleGAN-2 coupled with SimCLR/SimSiam/BYOL demonstrate the method's effectiveness, often outperforming the contemporary view-generation strategies such as image augmentations and using pre-generated synthetic-views.

**Audience:**

Yes

**Broader Impact Concerns:**

- While continuous sampling reduces disk IO, continuous sampling of high-resolution images from large generators is still GPU-intensive. The authors do study CO2 emission explicitly but I am afraid that any approach to scale up this method would be detrimental to environment.

- Synthetic-data biases – Latent perturbations may amplify biases present in the generator (e.g. over-representation of certain demographics) and downstream models might inherit those biases unnoticed.

**Claims And Evidence:**

Yes

**Requested Changes:**

I have following questions. They are not typical "requested changes" but any follow-up explanation/experiment would be helpful.

- Could the latent-space perturbation functions be constrained to specific Lie-group actions (e.g., translation, rotation, isotropic scaling) so that each learned direction corresponds to a single geometric transform on the image? If so, how would you enforce and verify that disentanglement?

- Positive-pair definitions can be application-dependent (e.g., background variance with foreground fixed). Can your framework target such feature-conditioned perturbations—changing class-irrelevant factors while preserving class-critical ones—and how do you label or discover which latent offsets achieve each effect?

- Beyond positives, could the same MI-based metric be inverted to sample hard negatives—perturbations that leave low-level appearance similar yet flip high-level semantics?

- The paper assumes a direct relationship between the L2 norm of a latent perturbations and the resulting MI shift in the pixel space. Can’t the perturbations not introduce changes that are can/will not be explained by the mutual information in pixel space? for e.g., how would account for latent perturbation that change illumination or texture in ways MI fails to capture?

- Given the probabilistic nature of the generator, do identical latent perturbation sometimes yield different MI shifts across stochastic draws? If so, how is this variability reflected in your ranking of “important” vs. “under-utilised” latents?

- When InfoNCE embeddings are already trained for invariance, does measuring MI in that embedding space risk underestimating meaningful pixel-space changes? Why not evaluate MI directly between image distributions (e.g., with MINE) to avoid embedding-induced blind spots?

**Strengths And Weaknesses:**

**Strengths**

I found the paper to be very well-written and easy to read. The authors did a good job at sharing useful insights regarding the impact of latent perturbations in MLVGMs. I appreciate the ablations on perturbations strengths and MI thresholds. The approach to generative views is very effective and the downstream results with contrastive learning indicate that.

**Weaknesses**

While the paper demonstrates good results for positive views generation via their perturbations of MLVGMs, I have a few concerns.

- The authors do not study an explicit mapping from norm size to types of semantic change in pixel space. There is thus interpretability gap for perturbation i.e., the optimisation treats the perturbation as “smallest L2-norm offset that lowers MI,” but because the direction is unconstrained, the same-sized perturbation may manifest as colour shifts, pose changes, or texture noise; hence magnitude alone does not map to a specific, repeatable semantic transformation, which limits the method’s utility for tasks that demand controlled edits.

- There has been some work on identifying latent directions and consequently leveraging them for interpretable, controlled generation in GAN which is very relevant to this work [1, 2]. It would be interesting to compare these methods for the same SSCRL tasks?


- Continuous Sampling is claimed to circumvent storage and sample diversity issue. While authors show that online generation in continuous sampling is on par or better than dataloading. It adds other issues for e.g.,
1. is final accuracy of a deterministic contrastive downstream model reproducible given on-the-fly stochasticity in training?
2. The latency also depends on the generative model and with bigger model it would be hard to scale it.
3. The generative model will occupy precious gpu compute that can be in turn used for increased batch size for contrastive model training. Have authors looked into this aspect?


References:
1. GANSpace: Discovering Interpretable GAN Controls [Erik Härkönen, Aaron Hertzmann, Jaakko Lehtinen, Sylvain Paris]
2. Optimal Positive Generation via Latent Transformation for Contrastive Learning [Yinqi Li, Hong Chang, Bingpeng MA, Shiguang Shan, Xilin Chen]

---

> ### Author Response · Authors · 2025-07-05
>
> Dear Reviewer jyeL,
>
> Thank you for taking the time to read our work and for your insightful feedback. In the following, we answer to each of your query, hoping to address your concerns and providing further explanations. Please be aware that, due to space constraints, we engage the discussion on the "requested changes" in a separate comment.
>
> **On studying direction for controllable edits, Ganspace and experiments**
>
> One key observation of our work lies in the fact that different latent spaces map to distinct semantic concepts, even when relying on perturbation magnitude alone. For example, latent $4$ of the BigBiGan model consistently encodes color information, with no observable entanglement with pose or texture. In other terms, we outline that MLVGMs already possess some kind of rough but unsupervised disentenglement between latent spaces, which we call "coarse-to-fine", and is independent from latent directions. As experimentally shown, such partial disentenglement is already extremely useful for certain tasks, where fine-grained controlled edits are not necessary (e.g. SSCRL).
>
> Undoubtedly, understanding how to control specific directions within each latent space may be a promising research contribution for high controllable image alterations (e.g. controlling which color to obtain in BigBiGan latent $4$), with even more applications across different tasks. Nonetheless, since the starting hypothesis of SSCRL is that of an unknown downstream task for the learned representations, it is not possible to determine a priori which particular semantic changes would result in stronger visual representations. Consequently, controlled generations, such as those in GANSpace [1], cannot be fully leveraged by view generation techniques for SSCRL. In fact, methods like [2] act solely on perturbation magnitudes, learning stronger and stronger image alterations. In Tables 2-3 of the main paper, this method is compared to ours, and we denote it with the term ``learned''. For more clarity, we have included direct citations in the revised version of the paper.
>
> [1] GANSpace: Discovering Interpretable GAN Controls [Erik Härkönen, Aaron Hertzmann, Jaakko Lehtinen, Sylvain Paris]
>
> [2] Optimal Positive Generation via Latent Transformation for Contrastive Learning [Yinqi Li, Hong Chang, Bingpeng MA, Shiguang Shan, Xilin Chen]
>
> **On Continuous Sampling**
>
> 1) During training, we fixed seeds based on device rank and iteration number, in order to generate different batches at each iteration and for each device, but allowing full reproducibility across runs.
> 2) Latency (or sampling speed) mainly depends on model type, more than model dimension. For example, GigaGAN [3] has 1B parameters but allows to generate 512px images in 0.13s. Therefore, we expect latency to be scalable for any fast-sampling model, even very large ones, while some limitations remain for Diffusion or Autoregressive models, which are notoriously slower at inference.
> 3) The generative model is used only in inference mode (no gradient computation). Therefore, the impact on GPU memory remains limited for small/medium models. For large models, one can consider using half precision or similar techniques to limit the occupied memory. Alternatively, for large batch sizes one may perform a loading (to GPU) and unloading (to CPU) of the model at each step, occupying GPU memory only during batch generation, not model training. We tested this alterative on BigBiGAN for a batch size of 256, and measured an average time per epoch of 249.59 seconds, which is greater than keeping the model on GPU the whole time, but still reduces timing w.r.t. standard torch loader (see Figure 5 of the main paper, left).
>
> More in general, we present Continuous Sampling (CS) as a valid alternative for increasing data diversity and maintain a competitive training time. We agree that, at present time, this technique may not be easily applicable to any model architecture (e.g. Diffusion models) or model size (depending on GPU availability). However, due to the fast progress of research and hardware technologies, these problems will inevitably attenuate in the next future, as more powerful models become faster and GPU capabilities increase, making CS more and more relevant w.r.t. offline loading techniques. For this reason, we believe our contribution remains valid for present and future research in computer vision.
>
> [3] Kang, Minguk, et al. "Scaling up gans for text-to-image synthesis." Proceedings of the IEEE/CVF conference on computer vision and pattern recognition. 2023.

---

> ### Author Response · Authors · 2025-07-05
>
> **On Lie-group actions**
>
> From what we observed during our experiments, different transformations (include Lie-group actions) largely depend on the latent level and generative model. For instance, we observed that the first group of latents in StyleGan-2 allows for transformations such as rotations, zoom and scaling. However, while several works, like [4, 5], try to achieve controllable edits on StyleGan, they do not consider the multiple latent variables involved in the process, and they are not specifically tailored for Lie-group actions.
>
> [4] Alaluf, Yuval, et al. "Hyperstyle: Stylegan inversion with hypernetworks for real image editing." CVPR 2022.
>
> [5] Pehlivan, Hamza, et al. "Styleres: Transforming the residuals for real image editing with stylegan." CVPR 2023.
>
> **Obtaining specific feature transformations**
>
> One assumption behind SSCRL is that positive pairs must be application unaware, as discussed in [6]. Therefore, in our study we did not consider the proposed scenario, which may be however applicable to other contexts (e.g. data augmentation for supervised learning). In this respect, and more in general in order to enhance interpretability and control over image edits, different solutions may apply:
>
> I) Class-conditioned MLVGMs, such as StyleGan-XL [7] may be immediately employed to obtain countless augmentations, while ensuring to remain within in the same class.
>
> II) Another interesting option, which we are investigating, is the use of text-to-image MLVGMs such as GigaGAN [3]. More in detail we are studying how simple prompts (e.g. "rotate left", or "red background") impact on each latent level, with the final aim of enforcing specific transformation within one or more latent level.
>
> III) At last, to remain in the context of unconditioned generation, models like SODA [8] propose to specialize different latent levels to encode specific transformations, mixing a masking mechanism with a view generation pretext-task.
>
> [6] What Makes for Good Views for Contrastive Learning? Tian et Al., NeurIPS 2020
>
> [7] Sauer, Axel, et al. "Stylegan-xl: Scaling stylegan to large diverse datasets." ACM SIGGRAPH 2022.
>
> [3] Kang, Minguk, et al. "Scaling up gans for text-to-image synthesis." CVPR 2023.
>
> [8] Hudson, Drew A., et al. "Soda: Bottleneck diffusion models for representation learning." CVPR 2024.
>
> **Sampling hard-negatives**
>
> Yes, similarly to how we sample positives, but with opposite logic. In this case we can employ perturbations that do not alter the last latent layers (color, texture, details), while slightly alter the first ones, typically encoding class-relevant aspects such as subject shape. However, since some frameworks do not include negatives (e.g. SimSiam), we opted for positives generation in our work.
>
> **On latent perturbations and MI shift**
>
> We are grateful to the reviewer for this very interesting observation, and we included the following considerations in the Limitations paragraph of our paper.
>
> As shown in Table 1, a direct relationship between latent space perturbations and MI shifts exists, even tough such mapping is not perfect. Theoretically, Mutual Information is invariant under invertible transformations. As a consequence, some small perturbations (e.g. a constant scaling to each pixel), will be approximated as a 0 shift by our method. Nonetheless, as observable in Table 1, the differences between each latent space are usually significant, especially for the last layers (where small, invertible transformations are more easily encoded). Therefore, it is reasonable to assume that such small approximations do not invalidate our findings.
>
> **On the probabilistic nature of the generator**
>
> Yes, the same L2 norm will often lead to slightly different MI shifts. In our study, we consider this aspect and statistically overcome it by sampling a large number of perturbations. In Table 1, we then report the average perturbation, along with the encountered standard deviation.
>
> **On others MI estimators (e.g. MINE)**
>
> As detailed in Equation 3 of the main paper, InfoNCE loss poses a lower bound on Mutual Information between pixel-space views. Nonetheless, the proposed method and its training dynamics (see related paragraph in Section 3.1 of the new paper version) allow to start from a situation of maximized Mutual Information (no perturbation applied), and slowly decrease it during training. This procedure enables slow convergence, somewhat counteracting the risk of underestimating meaningful pixel-space transformations.
>
> On the other hand, while other MI estimators (e.g. MINE) could be applied to measure MI between image distribution, doing so would imply a very complex procedure. In fact, MINE requires specific training for each couple of distributions, which would not only require training each latent space, but also for different perturbation magnitudes, since these will generate a different distribution.
>
> **Broader Impact:** Thank you, we considered your comments in the Conclusions.

---

### Author Response · Authors · 2025-07-05
**General Comment to All Reviewers**

Dear Reviewers,

Thank you for taking the time to read our work and for giving appropriate feedback. Following your suggestions, we uploaded an improved version of the manuscript (all changes are written in **red**).

We also addressed each of your concerns in separate comments.

---

### Decision · Action_Editor_GWKm · 2025-08-28

**Recommendation:** Accept as is

**Additional Comments:**

N/A

**Audience:**

Yes

**Audience Explanation:**

This paper presents an image generation method by using multiple latent variables, which is of great interest to researchers and practioniers in this field.

**Claims And Evidence:**

Yes

**Claims Explanation:**

All the issues raised by the reviewers have been addressed well by the authors. All reviewers express positive feedback regarding the acceptance of this paper.

All the claims made in this paper are supported by accurate, convincing, and clear evidence.